# A unified computational framework for single-cell data integration with optimal transport

Kai Cao [1,2,5], Qiyu Gong[3,5], Yiguang Hong [4] ✉ & Lin Wan [1,2] ✉

Single-cell data integration can provide a comprehensive molecular view of cells. However, how to integrate heterogeneous single-cell multi-omics as well as spatially resolved transcriptomic data remains a major challenge. Here we introduce uniPort, a unified single-cell data integration framework that combines a coupled variational autoencoder (coupled-VAE) and minibatch unbalanced optimal transport (Minibatch-UOT). It leverages both highly variable common and dataset-specific genes for integration to handle the heterogeneity across datasets, and it is scalable to large-scale datasets. uniPort jointly embeds heterogeneous single-cell multi-omics datasets into a shared latent space. It can further construct a reference atlas for gene imputation across datasets. Meanwhile, uniPort provides a flexible label transfer framework to deconvolute heterogeneous spatial transcriptomic data using an optimal transport plan, instead of embedding latent space. We demonstrate the capability of uniPort by applying it to integrate a variety of datasets, including single-cell transcriptomics, chromatin accessibility, and spatially resolved transcriptomic data.

The latest developments in high-throughput single-cell multi-omics sequencing technologies, e.g., single-cell RNA-sequencing (scRNA) and single-cell Assay for Transposase-Accessible Chromatin using sequencing (scATAC), enable comprehensive studies of heterogeneous cell populations that make up tissues, the dynamics of developmental processes, and the underlying regulatory mechanisms that control cellular functions. The computational integration of single-cell datasets is drawing heavy attention toward making advancements in machine learning and data science[1–3].

Among existing single-cell integration methods, tremendous efforts[4–7] have been devoted to integrating multiple datasets simultaneously profiled from the same cells (e.g., paired-cell datasets generated by the cellular indexing of transcriptomes and epitopes by sequencing (CITE-seq[8]). However, these paired datasets are technically challenging and costly to obtain. Therefore, a vast number of integrative methods have been developed for data profiled from different cells taken from the same, or similar, populations. For example, the celebrated platform Seurat[9] projected feature space into a common subspace using canonical correlation analysis (CCA), which maximizes inter-dataset correlation. LIGER[10] and DC3[11] employed non-negative matrix factorization to find the shared low-dimension factors of the common features to match single-cell omics datasets. Harmony[12] iterated between maximum diversity clustering and a mixture model-based linear batch correction, providing a latent space in which batch effects are removed. However, these methods rely on linear operation, thus lacking the ability to handle nonlinear deformations across cellular modalities. In addition, they only leverage filtered common genes, while ignoring the importance of dataset-specific genes for the identification of cell populations, which usually capture cell-type heterogeneity not present in common genes[13].

[1]LSC, NCMIS, Academy of Mathematics and Systems Science, Chinese Academy of Sciences, Beijing, China. [2]School of Mathematical Sciences, University of Chinese Academy of Sciences, Beijing, China. [3]Shanghai Institute of Immunology, Faculty of Basic Medicine, Shanghai Jiao Tong University School of Medicine, Shanghai, China. [4]Department of Control Science and Engineering, Tongji University, Shanghai, China. [5]These authors contributed equally: Kai Cao, Qiyu Gong. ✉e-mail: yghong@iss.ac.cn; lwan@amss.ac.cn

To address these shortcomings, manifold alignment methods are emerging and have achieved promising results in integrating single-cell multi-omics datasets[14–17]. However, manifold alignment methods are limited by relatively high computational complexity, and they are not scalable to large-scale datasets.

With the development of deep learning, many autoencoder-based approaches have been proposed and demonstrated their power in data integration across modalities. However, most of them require paired datasets profiled from the same cells, such as DCCA[18] and Cobolt[19], to utilize cell-paring information. When cell-paring information is unavailable, the alternative is simultaneous training of different autoencoders and aligning cells across different modalities in a latent space. However, this option still makes computation a challenging exercise. Recently, an emerging number of methods have been developed to account for unpaired data. For example, methods like scDART[20] and cross-modal autoencoders[21] attempted to learn a latent space by autoencoders and align latent representations via kernel-based or discriminator-based discrepancy. However, these methods require global alignment which is often too restrictive for integrate heterogeneous cellular populations. In addition, the transfer learning-based methods were also developed to establish a source atlas via one modality for knowledge (e.g., cell labels) transfer to another modality by learning a modality-invariant latent space[22,23]. Although having achieved encouraging results, these methods are restricted to using source modality with annotated cell labels.

Recently published methods for single-cell genomics integration such as scMC[24] and SCALEX[25] showed state-of-the-art performance on batch effect correction of one modality, but they have not been benchmarked on single-cell multi-omics data integration. GLUE[26], another state-of-the-art method for single-cell multi-omics (e.g., scA-TAC, scRNA) integration and integrative regulatory inference, based its development on advanced graph autoencoders. Meanwhile, many other methods are proposed for integrative analysis of spatial transcriptomics (ST) and scRNA data. Among these methods, gimVI[27] and Tangram[28] achieved the most advanced performance[29]. However, to the best of our knowledge, no method has been developed for a unified integration of single-cell multi-omics as well as spatially resolved transcriptomic data.

To address this gap, we herein advance the field by developing uniPort, an accurate, robust, and efficient computational platform for integrating heterogeneous single-cell datasets with optimal transport (OT). To overcome the limitation thwarting conventional VAE for single-cell heterogeneous and/or unpaired data integration, we propose a unified computational framework by combining a coupled variational autoencoder (coupled-VAE) and Minibatch Unbalanced OT (Minibatch-UOT)[30] (Fig. 1). This framework allows leveraging both highly variable common and dataset-specific genes for integration in order to handle the heterogeneity across datasets. Experimental results show that uniPort can accurately and robustly integrates scATAC and scRNA datasets profiled from peripheral blood mononuclear cells (PBMC) and mouse spleen. It can also accurately impute unmeasured spatially resolved multiplexed error robust fluorescence in situ hybridization (MERFISH)[31] genes through scRNA data. Moreover, with an output OT plan, we demonstrate that uniPort can accurately decipher canonical structures of the mouse brain and assist in locating tertiary lymphoid

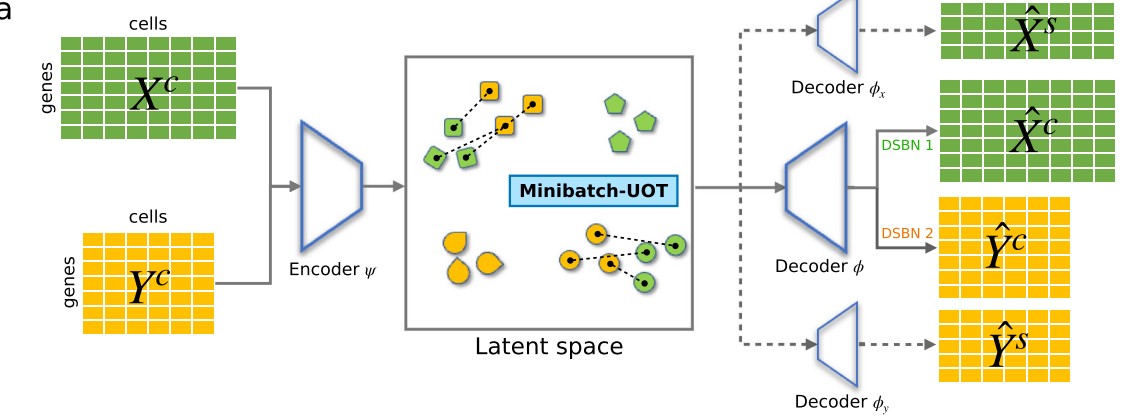

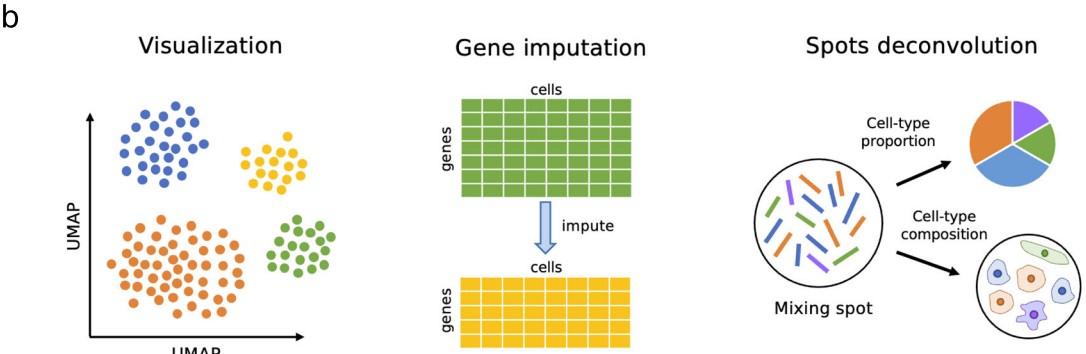

**Fig. 1 | Overview of uniPort algorithm.** uniPort integrates single-cell data by combining a coupled-VAE and Minibatch-UOT. uniPort takes as input a highly variable common gene set of single-cell datasets across different modalities or technologies. **a** uniPort projects input datasets into a cell-embedding latent space through a shared probabilistic encoder. Then uniPort minimizes a Minibatch-UOT loss between cell embeddings across different datasets. Finally, uniPort reconstructs two terms. The first consists of input datasets by a decoder with different DSBN layers. The second consists of highly variable gene sets corresponding to each dataset by dataset-specific decoders. **b** uniPort outputs a shared latent space and an optimal transport plan that can be used for downstream analysis, such as visualization, gene imputation and spots deconvolution.

structures (TLS) in the breast cancer region, as well as reveal cancer heterogeneity in microarray-based spatial data.

## Results

### uniPort embeds and integrates datasets by coupled-VAE and Minibatch-UOT

As input, uniPort takes diverse and heterogeneous single-cell datasets across different modalities or technologies. uniPort is based on a coupled variational auto-encoder (coupled-VAE) and employs a dataset-free encoder to project highly variable common gene sets of different datasets into a generalized cell-embedding latent space. Then uniPort reconstructs two terms. One is input by a dataset-free decoder with dataset-specific batch normalization (DSBN)[25,32] layers. The other is a highly variable gene set through a dataset-specific decoder corresponding to each dataset (Fig. 1 and Supplementary Fig. 1). Some overlapping genes are often found between the two terms as some common genes are also highly variable in each dataset. However, with slight abuse of 'specific', we still name the second term a dataset-specific gene set in the following context. During integration, uniPort minimizes a Minibatch-UOT loss between cell embeddings in the latent space from different datasets. It is necessary to introduce the loss as it feeds back a gradient to the encoder to achieve a better alignment result, especially when dataset-specific decoders are considered that increase the heterogeneity across datasets in the latent space. Meanwhile, the minibatch strategy substantially improves the computational efficiency of OT, making it scalable to large datasets, and the unbalanced OT makes it more suitable for heterogeneous data integration.

We employed a series of scores to assess the performance of single-cell data integration. To quantify dataset mixing and cell-type separation, we computed two scores used by SCALEX: the Batch Entropy score[33] to evaluate the extent of mixing cells across datasets and the Silhouette coefficient[34] to evaluate the separation of biological distinctions. To benchmark annotation clustering accuracy, we adopted the adjusted rand index (ARI), the normalized mutual information (NMI), and the F1 scores using cell-type annotations (Methods). Then, for paired datasets, we employed the average fraction of samples closer than the true match (FOSCTTM)[14] to measure the preservation of cell–cell correspondence across datasets.

### uniPort integrates scATAC and scRNA data

We benchmarked uniPort against current state-of-the-art single-cell genomics integration methods[9,10,12,13,15,17,24–26,35] on one dataset of paired scATAC and scRNA (the paired PBMC dataset[36]) and two datasets of unpaired scATAC and scRNA (the microfluidic-based PBMC dataset[37] and the mouse spleen dataset[13]). We employed Uniform Manifold Approximation and Projection (UMAP)[38] to visualize the integration results.

We first applied uniPort to integrate the paired PBMC dataset (Fig. 2). The pairing information was only used for performance evaluation. We found that uniPort and GLUE achieved the best performance with comparable results (Fig. 2c–e). Specifically, uniPort achieved the highest Silhouette coefficient of 0.64, while GLUE had the second highest Silhouette coefficient of 0.621; uniPort had the second best average FOSCTTM of 0.0694, a total score of ARI, NMI and F1 of 2.321, and the third highest Batch Entropy score of 0.64, slightly below GLUE (average FOSCTTM of 0.0441, total score of 2.514 and Batch Entropy score of 0.677). Among all compared methods, uniPort, Seurat, Harmony, SCOT, and GLUE accurately integrated most cell types in two modalities (Fig. 2b and Supplementary Fig. 2).

In addition to integrating the paired PBMC dataset, we further evaluated uniPort on an unpaired microfluidic-based PBMC dataset (Supplementary Fig. 3). As a result, uniPort accurately integrated the scATAC and scRNA data with competitive performance comparable to

that of GLUE, MultiMAP and Harmony. For example, uniPort aligned most cell types well in two modalities (Supplementary Fig. 3a, b), demonstrating Silhouette coefficient and Batch Entropy score of 0.68 and 0.623 (Supplementary Fig. 3c, d), which were similar to GLUE (0.682 and 0.638, respectively), MultiMAP (0.648 and 0.623, respectively), and Harmony (0.636 and 0.626, respectively), but surpassing other compared methods.

We also tested uniPort on another unpaired scATAC and scRNA profiled from the mouse spleen dataset (Fig. 3). uniPort, scMC, Harmony, and Seurat achieved the highest performance. Specifically, uniPort achieved the highest Silhouette coefficient of 0.709, slightly higher than scMC (0.704), Harmony (0.704), and Seurat (0.699); Harmony ranked first in Batch Entropy score of 0.676, higher than uniPort (0.632) and Seurat (0.671); scMC had the highest total score of ARI, NMI and F1 of 2.466, while uniPort (2.436), Harmony (2.416), and Seurat (2.437) followed close behind with a total score for each higher than 2.4, slightly below that of scMC.

In summary, among all methods, uniPort performed favorably when compared with recently published state-of-the-art methods, showing accurate and robust results across both paired and unpaired datasets.

### uniPort performs unbalanced matching tasks of heterogeneous datasets

uniPort minimizes a Minibatch-UOT loss, which is suitable for unbalanced matching and provides a strong guarantee for heterogeneous data integration. To evaluate the performance of uniPort on heterogeneous data integration, we conducted two unbalanced matching tasks by removing some cell types from scATAC or scRNA of mouse spleen, separately. First, we removed "DC", "Granulocyte", "Macrophage" and "NK" types from scATAC data, while keeping scRNA data unchanged, and denoted the integration task as ATAC unbalanced matching ("UBM-ATAC"). Second, we removed the same cell types from scRNA data, while keeping scATAC data unchanged, and denoted the integration task as RNA unbalanced matching ("UBM-RNA"). For comparison, we also defined the integration of complete mouse spleen data as balanced matching ("BM").

uniPort accurately identified and separated the cells of "DC", "Granulocyte", "Macrophage" and "NK" from other cell types in the two unbalanced matching cases, while still aligning modality-shared cell types well (Fig. 4a, b). We compared uniPort with GLUE, Harmony, Seurat, MultiMAP, and scMC, all of which achieved high accurate performance on the "BM" task. Among all methods, only uniPort and Seurat achieved stable performance in all three cases (Fig. 4c, d). uniPort had the highest total score of 2.225 and Silhouette coefficient of 0.676, in the case of "UBM-ATAC", and the second highest total score of 2.191 and the third highest Silhouette coefficient of 0.688, in the case of "UBM-RNA". Therefore, compared with the case of "BM", uniPort is more robust than the other methods when heterogeneity is presented in the datasets.

### uniPort integrates MERFISH and scRNA data

We further considered the integration of ST and scRNA data. Two main types of ST sequencing technologies are high-plex RNA imaging-based and barcoding-based. High-plex RNA imaging-based spatial sequencing has the advantage of single-cell precision with greater depth, but it is restricted to partial measurement with lower coverage. To test the performance of uniPort over high-plex RNA imaging-based data, we applied uniPort to integrate MERFISH and scRNA data[31].

Among 155 genes in the MERFISH data, we used 153 common genes in both scRNA and MERFISH for integration. We applied UMAP to visualize the integration results of cell embeddings by uniPort, Harmony, Seurat, SCALEX, scVI, gimVI[27] and MultiMAP (Fig. 5a, b and Supplementary Fig. 4). As shown in the figures, uniPort and scVI

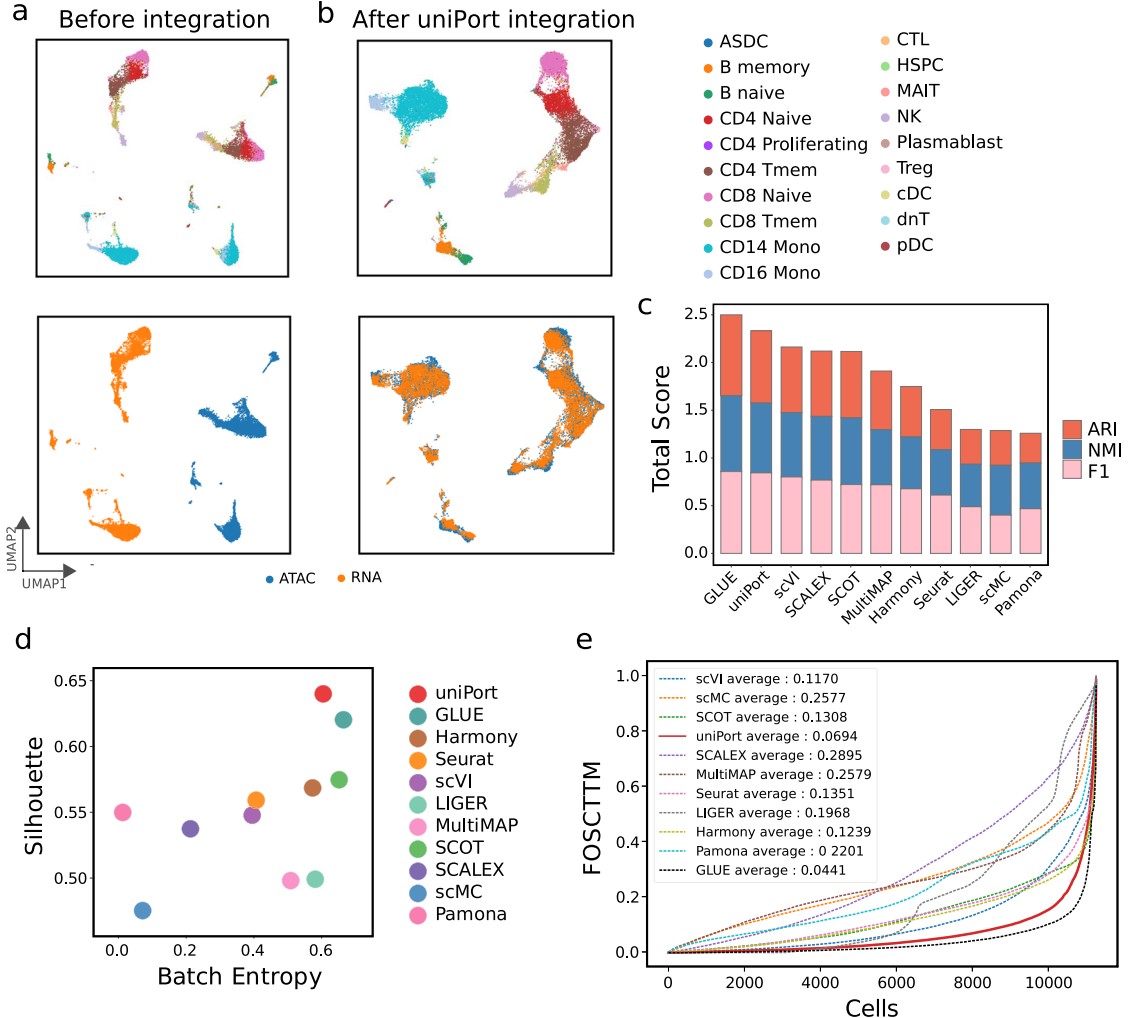

**Fig. 2 | uniPort integrates paired scATAC and scRNA of the PBMC data from 10×
Genomics. a** UMAP visualization of PBMC data before integration colored by omics
and cell annotations. **b** UMAP visualization of PBMC data after uniPort integration.
**c** Comparison of total scores of ARI, NMI and F1 of different methods.
**d** Comparison of Batch Entropy scores and Silhouette coefficients of different
methods. **e** Comparison of average FOSCTTM of different methods.

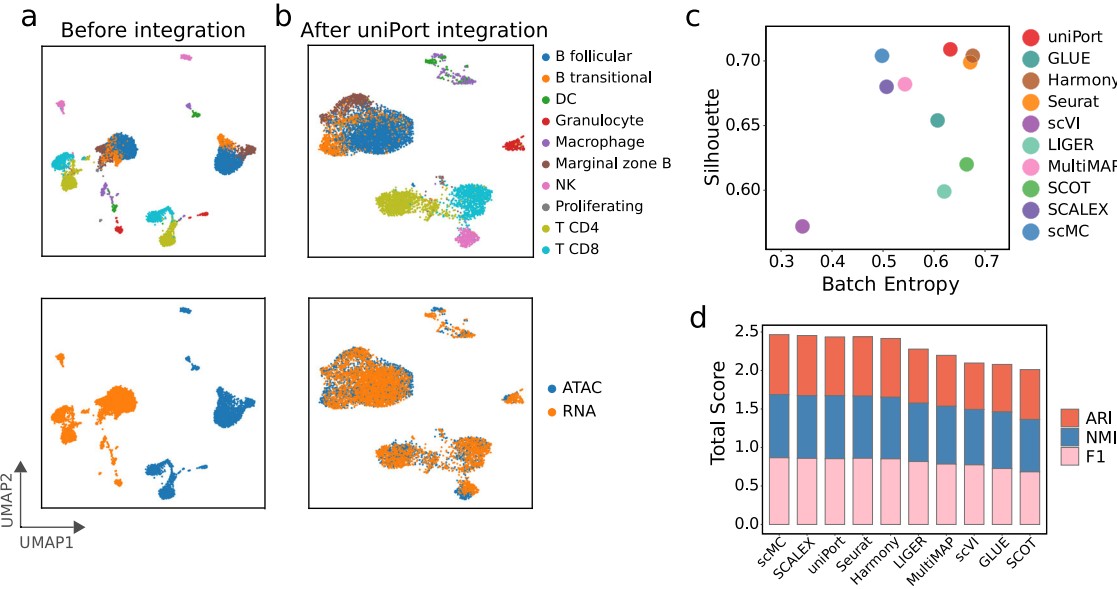

**Fig. 3 | uniPort integrates unpaired scATAC and scRNA of the mouse spleen
data. a** UMAP visualization of mouse spleen data before integration colored by
omics and cell annotations. **b** UMAP visualization of mouse spleen data after
uniPort integration. **c** Comparison of Batch Entropy scores and Silhouette coefficients of different methods. **d** Comparison of total scores of ARI, NMI and F1 of
different methods.

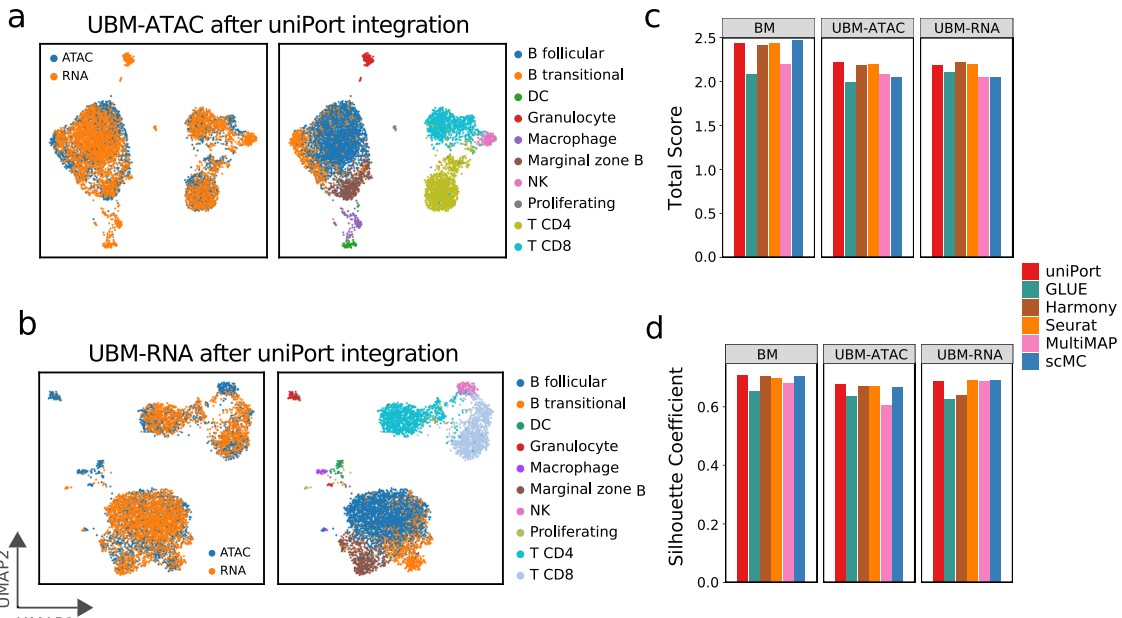

**Fig. 4 | uniPort integrates the cell-type unbalanced mouse spleen data. a** UMAP visualization of the case of "UBM-ATAC" after uniPort integration colored by omics and cell annotations. **b** UMAP visualization of the case of "UBM-RNA" after uniPort integration. **c** Comparison of total scores of ARI, NMI and F1 of different methods in the three cases. **d** Comparison of Batch Entropy scores and Silhouette coefficients of different methods in the three cases.

outperformed other methods in identifying and separating OD Immature cells from other cell types. Besides, uniPort accurately identified ependymal cells as a MERFISH-specific cell type and separated them from other cell types in the scRNA embeddings. We again benchmarked uniPort's integration performance against other methods[9,12,13,25,27,35] by the Silhouette coefficient and the total score (Fig. 5c, d). We found that uniPort outperformed other methods with the highest Silhouette coefficient of 0.706 and the highest total score of 2.404 for ARI, NMI and F1, while scVI achieved the second highest Silhouette coefficient of 0.688, and MultiMAP ranked second in the total score of 2.37.

### uniPort imputes genes for MERFISH data

uniPort trained an encoder network to project cells with common genes across datasets into a common cell-embedding latent space and a decoder network to reconstruct cells with both common and specific genes. Therefore, once the coupled-VAE is trained well, it can be regarded as a reference atlas, in turn allowing uniPort to impute both common and specific genes in one dataset through common genes in another dataset by the atlas (Supplementary Fig. 5). The imputed genes can be used to enhance the resolution of spatial transcriptomics[39,40].

To explore uniPort's ability for gene imputation, we followed the scheme of gimVI[27] to impute missing genes in MERFISH from scRNA. To be specific, we first randomly selected 80% (i.e., 122/153) genes in MERFISH as training genes and reserved the remaining 20% (i.e., 31/153) genes as testing genes. We repeated the above steps twelve times and obtained 12 training and testing gene sets. Afterwards, we trained the uniPort network with each training gene set, and then imputed the corresponding testing gene set. We compared our results with two state-of-the-art gene-imputing methods[29]: gimVI[27] and Tangram[28]. We applied uniPort, gimVI and Tangram to impute testing genes, and used UMAP to visualize both training and testing genes (Fig. 5e). With an imputation framework like that of gimVI, we also excepted imputed values of uniPort to carry gene-specific biases from scRNA genes[27]. Therefore, for performance evaluation, we followed gimVI and reported the median and average Spearman correlation coefficients (mSCC and aSCC), as well as the median and average Pearson correlation coefficients (mPCC and aPCC) over imputed and

ground truth testing genes. uniPort provided a significant improvement over the two compared methods on the MERFISH dataset. For example, uniPort separated different cell types in the UMAP visualization of imputed genes with a better result (Fig. 5e), and demonstrated the highest mSCC (0.259), aSCC (0.26), mPCC (0.249), and aPCC (0.294), significantly above those of gimVI (mSCC of 0.221, aSCC of 0.24, mPCC of 0.201, and aPCC of 0.242) and Tangram (mSCC of 0.188, aSCC of 0.206, mPCC of 0.202, and aPCC of 0.231) (Fig. 5f).

We further explored uniPort's ability for online imputation by training a model of scRNA and MERFISH data profiled from mouse #1, as a reference atlas, and imputing scRNA genes through MERFISH profiled from mouse #2. The result showed that the same genes in MERFISH and predicted scRNA were also significantly correlated, demonstrating the ability of uniPort to impute genes in an online manner (Supplementary Discussion 1 and Supplementary Fig. 6).

### uniPort deconvolutes synthetic STARmap data

Barcoding-based ST is more accessible to transcripts and achieves higher coverage, while it is limited to the mixing spots with lower resolution[41]. We next considered the deconvolution of barcoding-based ST data through transferring labels from scRNA data to spots. uniPort can provide an OT plan, which represents a cell-to-spot probabilistic correspondence between scRNA and ST data, allowing us to deconvolute the proportion of single-cell clusters for ST data according to cell annotations in scRNA data (Supplementary Method 2). Here, we first applied uniPort to deconvolute synthetic STARmap data[29]. To evulate performance, we benchmarked uniPort against two state-of-the-art cell-type deconvolution methods, Tangram[28] and SpaOTsc[42]. Tangram is a global matrix optimization method, which aims to find a mapping matrix from which to project scRNA-seq data to spots. SpaOTsc is an OT-based method that applies unbalanced and structured Gromov-Wasserstein OT to find a matching matrix between scRNA-seq data and spots.

We benchmarked the results of uniPort, Tangram, and SpaOTsc with four scores: Pearson correlation coefficient (PCC), structural similarity index (SSIM), root mean square error (RMSE), and Jensen-Shannon divergence (JSD) introduced by a recent paper that benchmarked spatial and single-cell transcriptomics integration methods[29].

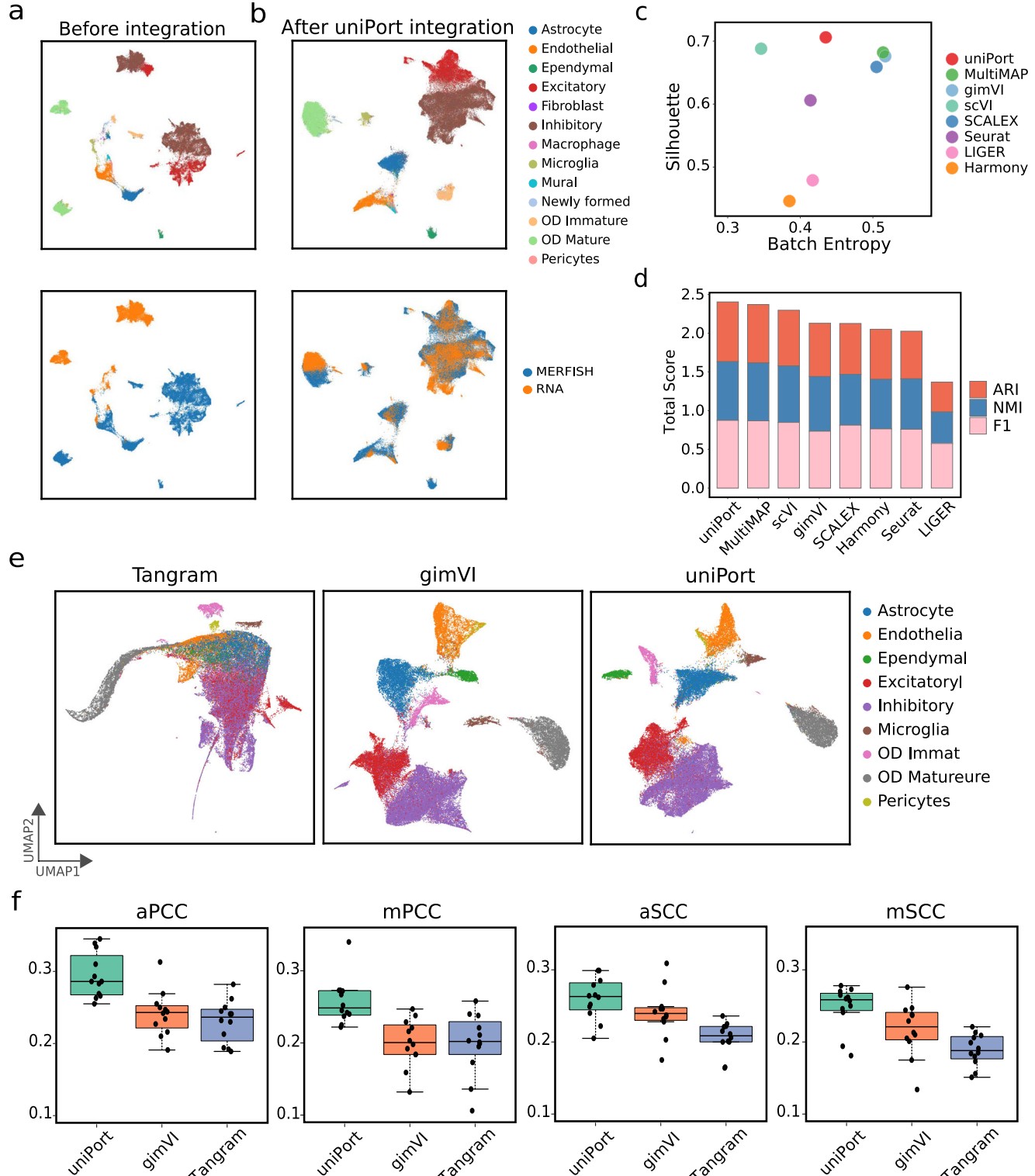

**Fig. 5 | uniPort imputes MERFISH genes through scRNA data. a** UMAP visualization of MERFISH and scRNA data before integration. **b** UMAP visualization of MERFISH and scRNA data after uniPort integration. **c** Comparison of Batch Entropy scores and Silhouette coefficients of different methods. **d** Comparison of total scores of ARI, NMI and F1 of different methods. **e** UMAP visualization of imputed MERFISH genes of Tangram, gimVI and uniPort. **f** Boxplots of average and median Pearson correlation coefficients (aPCC and mPCC) ($n = 12$, no statistical method was used to predetermine sample size), and average and median Spearman correlation coefficients (aSCC and mSCC) ($n = 12$) between real and imputed MERFISH genes. In the boxplots, the center line, box limits and whiskers denote the median, upper and lower quartiles and $1.5\times$ interquartile range, respectively.

Higher PCC and SSIM and lower RMSE and JSD, indicate better performance. We adopted the results of Tangram and SpaOTsc directly from the benchmarking paper[29]. Overall, uniPort performed competitively with the two methods: uniPort performed favorably with PCC of 0.449, RMSE of 0.157, and JSD of 0.569, which is below Tangram (PCC of 0.619, RMSE of 0.147, and JSD of 0.524) while above SpaOTsc (PCC of 0.409, RMSE of 0.197, and JSD of 0.573) (Supplementary Fig. 7).

## uniPort deciphers canonical structures of mouse brain

We then applied uniPort to deconvolute real-world barcoding-based ST examples. To estimate cell-type composition for each captured spot and decipher typical organizational structures, we first integrated the anterior slice of adult mouse brain ST data (10× Visium)[43].

As clear-cut boundaries exhibited, uniPort accurately reconstructed the well-structured layers and deconvoluted 28 cell types (Fig. 6a). The proportion and position of representative clusters, e.g., multiple cortical layers and region-specific cell types, are highly consistent with those of previous studies[43,44]. Despite the complexity of its anatomy, uniPort accurately remodeled and arranged the L2/3-L6 subclusters extending from the boundary to the central area (Fig. 6b). In addition, subpopulations of the L6 layer were separated with clear limits, revealing the sensitivity of our method to near-imperceptible signals. The non-neuronal neuroglia cells that provide neurons with support and protection, including astrocytes and oligodendrocytes[45], had corresponding sites paired with their marker genes (e.g., *Olig1* and *Olig2* of cluster Oligo; *Atp1b2* and *Slc1a2* of cluster Astro; *Dcn* and *Osr1* of cluster vascular and leptomeningeal cells (VLMCs)) (Supplementary Fig. 8). Moreover, VLMCs that form protective sections around the pia membranes of the brain also lie in the border of the slice in harmony with anatomical structures[46]. Therefore, our mapping is robust, as demonstrated by either expression of marker genes or anatomy of the brain, and can establish an agreement between gene expression-based clustering and anatomical annotation, providing a more thorough and comprehensive understanding than can be achieved through visual inspection.

## uniPort assists in locating TLS in breast cancer region

The genesis and progression of cancer are generally influenced by their association with the heterogeneous tumor microenvironment (TME)[47] for which ST can provide biological insights. To further demonstrate its flexible utility, we used uniPort to deconvolute spatial data of HER2-positive breast cancer, containing diffusely infiltrating cells that make it more difficult to deconvolute spots. As shown in Fig. 6c, nine main clusters were assigned on spatial images, primarily involving T cells and cancer epithelial cells. Moreover, we found that representative clusters scattered in their centralized enrichment region coincided with the area indicated by the expression of their marker genes (Fig. 6d). For example, T and B cells, which establish crucial adaptive immunity through protective immunological memory, were matched well with canonical marker genes, such as *CD3D* and *MS4A1* (Fig. 6d, e). Myeloid cells, as an innate part of the immune system, also displayed a distribution concordant with the expression of CD68[48]. Furthermore, cancer epithelial cells protruded along the invasive ductal carcinoma region, corresponding with the expression of *ERBB2* as well. Overall, the above results reach concordance between pathological annotation and data-motivated labeling.

Massive research has demonstrated that an increased infiltration of immune cells is highly related to favorable breast cancer prognosis[49]. TLS, a kind of ectopic lymph-like organ recently discovered at sites of tumor or inflammation, are considered as a prognostic and predictive factor for cancer patients. Although TLS are inhabited by multiple cell types, the major residents are T and B cells, implicating the TME by their joint colocalization[50]. Through decomposing cell-type proportion of each spot, we identified TLS signals via colocalization of T and B cells, rendering an identical expression intensity with T and B cells (Fig. 6f). In general, our approach can harmonize diverse modalities and cater to both high-resolution mapping and recognition of representative architectures across tissues and diseases.

## uniPort reveals cancer heterogeneity in microarray-based spatial data

The area of Visium-based ST data is limited to a 55 μm diameter for each captured spot, which reaches a moderate resolution that translates to 3–30 cells[41]. Latent integration challenges may arise, along with the decrease of spot resolution, as an increased mix of ingredients brings more noise. To examine the performance of uniPort in this case, we employed microarray-based ST data of pancreatic ductal adenocarcinoma (PDAC) tissues for integration, the diameter of which stretches for 100 μm[51]. Cell-type deconvolution was applied on 428 spots paired with 1926 single cells, measuring 19,736 genes respectively.

We decomposed 15 main clusters, which exhibit a discrete enrichment and complexity of both normal and tumor composition (Fig. 7a). In detail, normal pancreatic cell types were classified as ductal and acinar cells consistent with the results of previous studies[52], preserving dramatically different distributions and genetic characteristics against those of cancer cells. As for malignant pancreatic cells, we grouped them as cancer clone A and B clusters based on genetic differences. Once again, histological annotations of normal and cancerous regions were, overall, in line with their data-driven labels (Fig. 7b), and essential constituents of the TME were indicated by their marker genes (Fig. 7c, d).

To gain further insight into the heterogeneity of cancer subtypes, we confirmed their identity, accounting for the maximum proportion of each spot (Fig. 7e). Top enrichment KEGG pathways isolated them into distinct functional assemblies (Fig. 7f). Cancer clone A is suspected to be an invasive phenotype attributed to the high enrichment of platelet activation and leukocyte transendothelial migration pathway. Lumps of data have proved that platelets are closely related to a high risk of metastasis in patients with pancreatic cancer[53]. Furthermore, the proportion of hematogenous cells, including red blood cells (RBCs), T cells, and natural killer (NK) cells, in cancer clone A significantly increased (Fig. 7g), which is consistent with the results of functional analysis. Tight junction plays a critical regulatory role in the physiologic secretion of the pancreas, and its disruption contributes to the pathogenesis of progressive pancreatic cancer[54]. Furthermore, PI3K signaling can potentially modify the TME to dictate the outcome, which must be considered to have therapeutic opportunities for targeting PDAC[55]. All these beneficial signals were enriched in the cancer clone B region where endothelial cells showed a significant presence, suggesting a less malignant cluster in contrast with cancer clone A. In sum, then, our method can manipulate an extensive application spectrum of varying resolutions, revealing the subtle heterogeneous TME.

## Discussion

We introduce a unified deep learning method named uniPort for single-cell data integration and apply it to integrate transcriptomic, epigenomic, spatially resolved high-plex RNA imaging- and barcoding-based single-cell genomics. uniPort combines a coupled-VAE and Minibatch-UOT and leverages both highly variable common and dataset-specific genes for integration. It is a nonlinear method that projects all datasets into a common latent space and outputs their latent representations between datasets, enabling both visualization and downstream analysis.

Generally, uniPort tackles several computational challenges, starting with removing the constraint of paired cells required by other autoencoder-based models by the employment of Minibatch-UOT. Different from existing methods that only consider common genes across datasets, we also take advantage of genes unique to each dataset, typically capturing cell-type heterogeneity not present in common genes (Supplementary Discussion 2 and Supplementary Figs. 9, 10). Besides, uniPort shows its power and potential for gene imputation by constructing a reference atlas owing to the generalization ability of coupled-VAE. It is relevant to point out that uniPort can even impute unique genes in one dataset through common genes in another dataset without having to train from scratch. Moreover, uniPort can output an OT plan for downstream analysis, such as

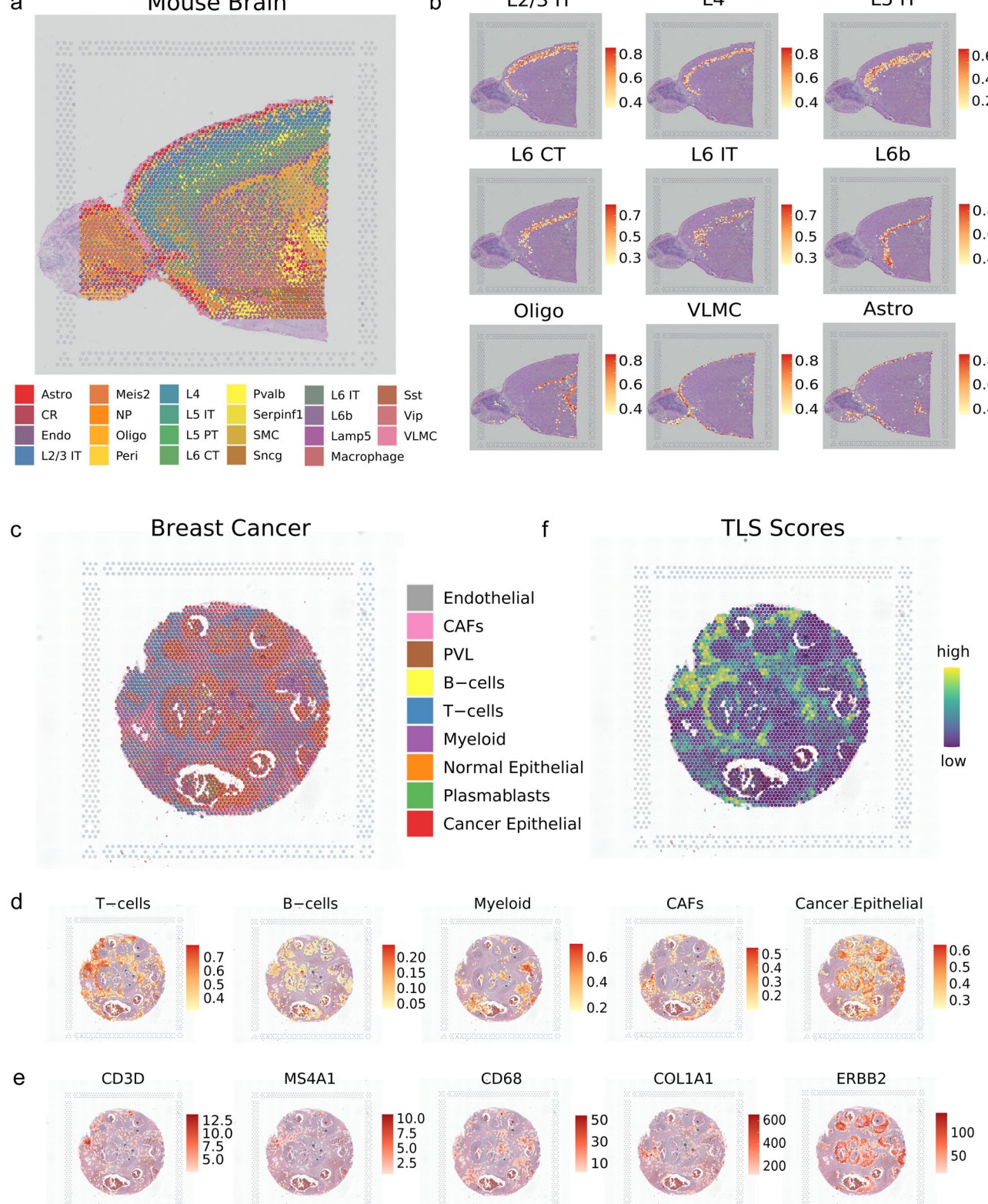

**Fig. 6 | uniPort identifies iconic structures in spatial transcriptomic data (10× Visium). a** Results of mapping spatial data to single-cell data using the optimal transport plan. Spatial scatter pie plot displays the well-structured cluster composition in adult mouse brain anterior slice. **b** Lists of canonical cerebral cortical neuron types with scaled proportion. **c** Spatial deconvolution result of the HER2-positive breast cancer. **d** Proportion of typical clusters in tumor microenvironment. **e** Expression of marker genes corresponding to clusters in **d**. **f** Tertiary Lymphoid Structure (TLS) scores inferred from summing the proportion of T cells and B cells together with their colocalization.

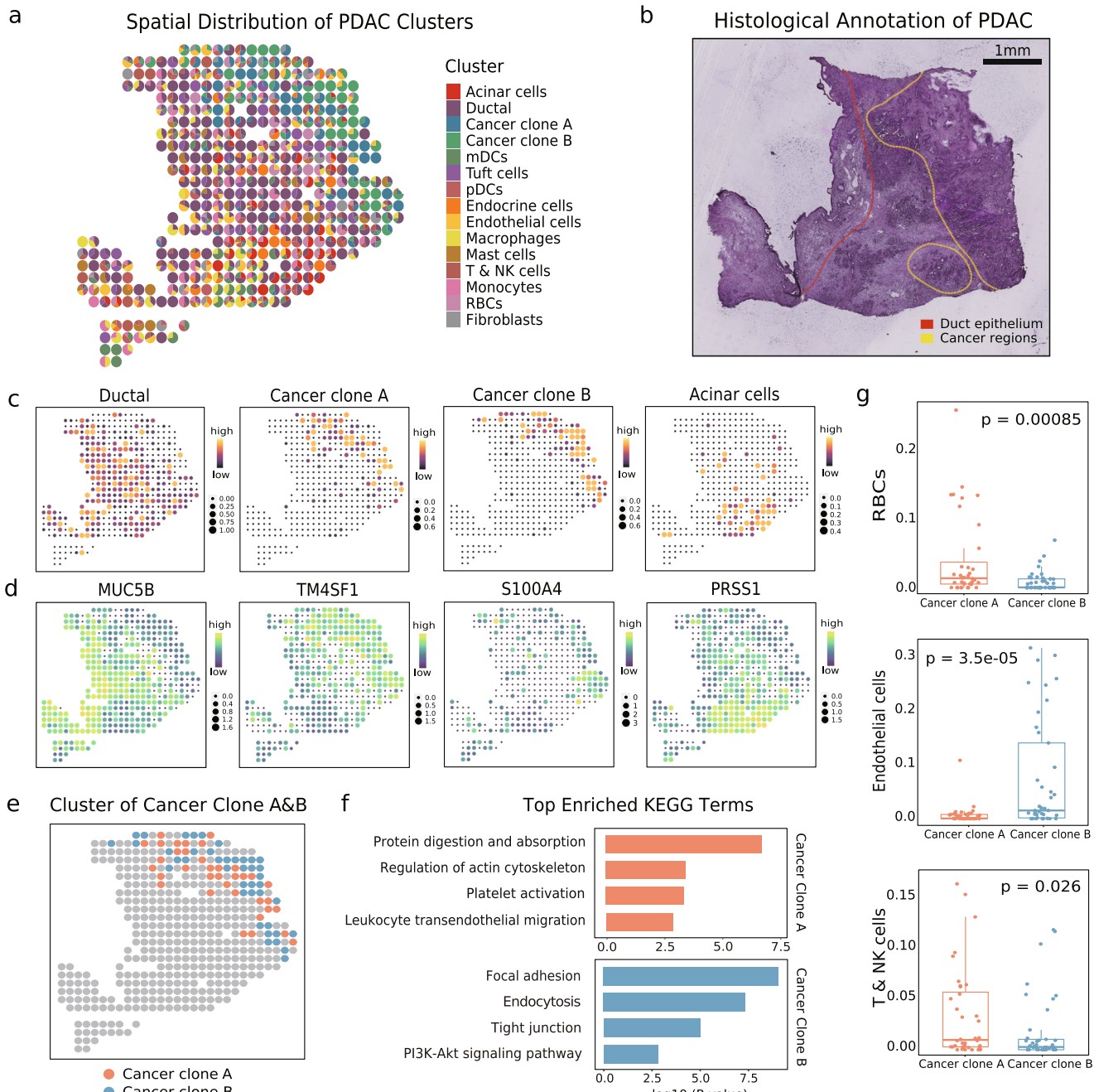

**Fig. 7 | uniPort identifies distinct cancer subtypes in microarray-based spatial data. a** Spatial deconvolution result of pancreatic ductal adenocarcinoma (PDAC). **b** Three manually segmented annotation of PDAC tumor cryosection on one ST slide. The red line circles the ductal epithelium region (left), and the yellow line circles the cancer region (right). **c** Proportion and distribution of typical clusters in PDAC. **d** Expression of marker genes corresponding to clusters in **c**. **e** Distribution of cancer clone subtypes. **f** Top enriched KEGG terms of distinct cancer subtypes. **g** Boxplots of significant differences of cluster composition between the cancer clone A ($n = 36$) and cancer clone B ($n = 41$) regions (two-sided $t$-test). In the boxplots, the center line, box limits and whiskers denote the median, upper and lower quartiles and $1.5 \times$ interquartile range, respectively.

flexible label transfer learning, for deconvolution of spatial heterogeneous data.

Although many mechanisms are involved, uniPort is still computationally efficient and scalable to integrate large-scale and heterogeneous datasets, which could be computationally prohibitive for other OT-based methods. To the best of our knowledge, prevalent OT-based methods for single-cell analysis are based on global optimal transport, e.g., SCOT[15] and Pamona[17] for single-cell multiomics data integration, SpaOTsc[42] and novoSpaRc[56] for spatial positions reconstruction, and Waddington-OT[57] for trajectory inference. Global optimal transport makes the computation very expensive. To resolve this drawback, our uniPort introduces a Minibatch-UOT into a VAE-based framework for single-cell genomics analysis, which only needs to solve a mini-batch transport plan at each iteration, thus significantly reducing the computational cost. Therefore, it is scalable to large datasets (Supplementary Discussion 3, Supplementary Figs. 11 and 12). Additionally, in general, our coupled-VAE and Minibatch-UOT-based uniPort is more accurate than other OT-based methods, such as SCOT, Pamona, and SpaOTsc in different tasks.

Given that our integration of scATAC is based on gene activity score, we also tested uniPort's performance when gene activity score is calculated in different approaches. We employed two methods to form the gene activity score introduced by Signac[58] and MAESTRO[37], respectively. Signac defines gene activity score for scATAC as the read count in the gene body and promoter region. MAESTRO calculates gene activity score as a weighted sum of nearby cis-regulatory elements (REs), where the weight is an exponentially decreasing function of distances of REs and target genes. The integration result showed that uniPort achieved better performance on the gene activity score by MAESTRO with all evaluated scores higher than that by Signac (Supplementary Fig. 13), which demonstrated the importance of modeling the gene activity score. It is worth noting that, GLUE[26], which is developed based on a sophisticated knowledge-based graph that explicitly and accurately models regulatory signals of scATAC, instead of using gene activity score, provides an important technique for analyzing scATAC data.

We demonstrate that uniPort consistently performs favorably when compared with recently published state-of-the-art methods, and successfully deconvolutes spatial heterogeneous data using the OT plan. With the rapid development of paired datasets and various heterogeneous modalities, we also demonstrate the generalizability of uniPort to other types of single-cell data by integrating paired datasets using CITE-seq data[6,8] and SNARE-seq data[59] (Supplementary Discussion 4, Supplementary Figs. 14, 15) or datasets without aligned common genes (Supplementary Discussion 5, Supplementary Fig. 16b). With no inherent reliance on any prior information, our framework offers the flexibility necessary to match prior information, e.g., cell-type annotations or cell-cell correspondence, when available (Supplementary Method 3, Supplementary Fig. 16c). We will keep updating and improving the framework in anticipation that uniPort will find a wide range of applications in the area of integrative single-cell multiomics data analysis.

## Methods
### uniPort framework

uniPort inputs each dataset though a coupled variational autoencoder (coupled-VAE) framework and learns its $K$-dimensional features. Given a sample $\mathbf{x}$, the corresponding $K$-dimensional latent vector $\mathbf{z}$ can be obtained by a variational posterior $p(\mathbf{z}|\mathbf{x})$ approximated by a probabilistic encoder $\psi(\mathbf{z}|\mathbf{x})$. Conversely, a probabilistic decoder $\phi(\mathbf{x}|\mathbf{z})$ produces a distribution over the possible corresponding values of $\mathbf{x}$ given $\mathbf{z}$. The coupled-VAE jointly learns $\psi$ and $\phi$ to maximize the evidence lower bound (ELBO) with a balanced parameter $\lambda$:

$$\mathcal{L}_{\text{ELBO}} = \mathbb{E}_{\psi(\mathbf{z}|\mathbf{x})}[\log \phi(\mathbf{x}|\mathbf{z})] - \lambda D_{\text{KL}}(\psi(\mathbf{z}|\mathbf{x}) \| p(\mathbf{z})). \tag{1}$$

The ELBO consists of a reconstruction term that encourages the output data to be similar to the input data, in addition to a Kullback-Leibeler divergence regularization term which regularizes the variational posterior to follow the prior distribution $p(\mathbf{z})$. We set the prior distribution to be the centered isotropic multivariate Gaussian $p(\mathbf{z}) = \mathcal{N}(\mathbf{z}; \mathbf{0}, \mathbf{I})$ and the variational posterior to be a multivariate Gaussian with a diagonal covariance structure $\psi(\mathbf{z}|\mathbf{x}) = \mathcal{N}(\mathbf{z}; \boldsymbol{\mu}, \boldsymbol{\sigma}^2 \mathbf{I})$, where the mean vector $\boldsymbol{\mu}$ and standard deviation vector $\boldsymbol{\sigma}$ are outputs of the encoder. Then, the latent vector $\mathbf{z}$ can be obtained through reparameterization by $\mathbf{z} = \boldsymbol{\mu} + \boldsymbol{\sigma} \times \nu$, where $\nu$ is sampled from $\mathcal{N}(\mathbf{0}, \mathbf{I})$.

Here, we formulate our method for the case of two datasets, while it can be easily generated for cases of multiple single-cell datasets. Suppose there are two single-cell datasets $\mathbf{X} = \{\mathbf{x}_1, \ldots, \mathbf{x}_{n_x}\}$ with $n_x$ cells and $d_x$ genes, and $\mathbf{Y} = \{\mathbf{y}_1, \ldots, \mathbf{y}_{n_y}\}$ with $n_y$ cells and $d_y$ genes. We first select the top $k$ highly variable common genes in both datasets to form $\mathbf{X}^c$ and $\mathbf{Y}^c$, and the top $k_x$ and $k_y$ highly variable genes in $\mathbf{X}$ and $\mathbf{Y}$ to form $\mathbf{X}^s$ and $\mathbf{Y}^s$, individually. We project both $\mathbf{X}^c$ and $\mathbf{Y}^c$ into a generalized cell-embedding latent space using a dataset-free probabilistic encoder

$\psi$ and a decoder $\phi$ with two Dataset-Specific Batch Normalization (DSBN) layers[32] corresponding to two modalities. Then, to leverage the information of dataset-specific highly variable genes, we also introduce two decoders, $\phi_x$ and $\phi_y$, to reconstruct $\mathbf{X}^s$ and $\mathbf{Y}^s$ from the latent variables, as well. Overall, the modified ELBO* loss for coupled-VAE is given by

$$\mathcal{L}_{\text{ELBO}^*} = \mathbb{E}_{\psi(\mathbf{z}|\mathbf{x})}\left\{ \log \phi(\mathbf{x}|\mathbf{z}) + \lambda_s \left( \log \phi_x(\mathbf{x}^s|\mathbf{z}) + \log \phi_y(\mathbf{y}^s|\mathbf{z}) \right) \right\} - \lambda D_{\text{KL}}(\psi(\mathbf{z}|\mathbf{x}) \| p(\mathbf{z})), \tag{2}$$

where $\mathbf{x} \in \mathbf{X}^c \cup \mathbf{Y}^c$, $\mathbf{x}^s \in \mathbf{X}^s$, $\mathbf{y}^s \in \mathbf{Y}^s$, and $\lambda_s, \lambda$ are balanced parameters. To better integrate heterogeneous single-cell datasets in the latent space, we design an alignment term for different datasets using Minibatch Unbalanced Optimal Transport (Minibatch-UOT)[30]. uniPort computes the Minibatch-UOT loss between datasets $\mathbf{X}$ and $\mathbf{Y}$ described as follows. For two $K$-dimensional Gaussian distributions $p_{x_i} = \mathcal{N}(\boldsymbol{\mu}_{x_i}, \boldsymbol{\sigma}_{x_i}^2 \mathbf{I})$ and $p_{y_j} = \mathcal{N}(\boldsymbol{\mu}_{y_j}, \boldsymbol{\sigma}_{y_j}^2 \mathbf{I})$ corresponding to cell $\mathbf{x}_i$ and $\mathbf{y}_j$, where $\boldsymbol{\mu}_{x_i}, \boldsymbol{\mu}_{y_j} \in \mathbb{R}^K$ and $\boldsymbol{\sigma}_{x_i}, \boldsymbol{\sigma}_{y_j} \in \mathbb{R}_+^K$ represent the output mean and standard deviation vectors by encoder $\psi$, respectively, the Minibatch-UOT cost between cell $\mathbf{x}_i$ and $\mathbf{y}_j$ is defined as (Supplementary Method 1):

$$\mathbf{C}_{ij} = \| \boldsymbol{\mu}_{x_i} - \boldsymbol{\mu}_{y_j} \|^2 + \| \boldsymbol{\sigma}_{x_i} - \boldsymbol{\sigma}_{y_j} \|^2. \tag{3}$$

We then compute the following optimal Minibatch-UOT plan[30] with batch size $B_x$ and $B_y$:

$$\mathbf{T}^* = \underset{\mathbf{T} \in \mathbb{R}_+^{B_x \times B_y}}{\arg\min} < \mathbf{C}, \mathbf{T} > - \epsilon H(\mathbf{T}) + \rho D_{KL}(\mathbf{T}\mathbf{1}_{B_y} \| \mathbf{a}) + \rho D_{KL}(\mathbf{T}^\top \mathbf{1}_{B_x} \| \mathbf{b}), \tag{4}$$

where $< \mathbf{C}, \mathbf{T} > = \sum_{i,j} \mathbf{C}_{ij} \mathbf{T}_{ij}$, entropy regularization term $H(\mathbf{T}) = -\sum_{i,j} \mathbf{T}_{ij}(\log \mathbf{T}_{ij} - 1)$, and $\mathbf{a} = \frac{1}{B_x} \mathbf{1}_{B_x}$, $\mathbf{b} = \frac{1}{B_y} \mathbf{1}_{B_y}$. $\epsilon$ and $\rho$ are balanced parameters. We set $\rho = 1$ for all experiments in this paper, but uniPort is robust to different choices of $\rho$ (Supplementary Fig. 17c). Eq. (4) is a strictly convex optimization problem and can be solved via an efficient inexact proximal point method (IPOT)[60] as

$$\boldsymbol{\alpha}^{(l+1)} = \left( \frac{\mathbf{a}}{\mathbf{G}\boldsymbol{\beta}^{(l)}} \right)^{\frac{\rho}{\rho+\epsilon}}, \boldsymbol{\beta}^{(l+1)} = \left( \frac{\mathbf{b}}{\mathbf{G}^\top \boldsymbol{\alpha}^{(l)}} \right)^{\frac{\rho}{\rho+\epsilon}}, \tag{5}$$

starting from $\boldsymbol{\beta}^{(0)} = \frac{1}{B_x} \mathbf{1}_{B_x}$, where $\mathbf{G}_{ij} = \mathbf{T}_{ij}^{(l)} e^{-\mathbf{C}_{ij}/\epsilon}$. The optimal Minibatch-UOT plan $\mathbf{T}_{ij}^* = \alpha_i \mathbf{G}_{ij} \beta_j$. Therefore, the Minibatch-UOT loss is given by

$$\mathcal{L}_{\text{UOT}^*} = \sum_{i,j} \mathbf{C}_{ij} \times \mathbf{T}_{ij}^*. \tag{6}$$

The total loss function minimized by uniPort is formulated as

$$\mathcal{L}_{\text{uniPort}} = -\mathcal{L}_{\text{ELBO}^*} + \gamma \mathcal{L}_{\text{UOT}^*}. \tag{7}$$

Once $\mathbf{T}^*$ is obtained, we provide users an option to output an OT plan $\mathbf{T}$ for tasks which need cell-to-cell probabilistic correspondence. Specifically, we first initialize $\mathbf{T} = \frac{1}{n_x n_y} \mathbf{1}_{n_x n_y}$. After each calculation of optimal Minibatch-UOT plan $\mathbf{T}^*$, we update $\mathbf{T}$ by replacing rows and columns by $\mathbf{T}^*$ which is sampled by the minibatch strategy. Note that storing a dense matrix $\mathbf{T}$ needs more memory. Therefore, computing capacity with respect to memory should be ascertained before using this option. We also provide an option for user-guided sample weights if cells are not uniformly matched. In this case, we set vectors $\mathbf{a}$ and $\mathbf{b}$ in Minibatch-UOT as weighted vectors specified by users, instead of uniform vectors (Supplementary Method 4).

## uniPort algorithm

uniPort integrates two single-cell datasets $\mathbf{X} = \{\mathbf{x}_1, \ldots, \mathbf{x}_{n_x}\}$ and $\mathbf{Y} = \{\mathbf{y}_1, \ldots, \mathbf{y}_{n_y}\}$.

**Input:** the top $k$ highly variable common genes in $\mathbf{X}$ and $\mathbf{Y}$, formulated as $\mathbf{X}^c$ and $\mathbf{Y}^c$, top $k_x$ and $k_y$ highly variable genes in $\mathbf{X}$ and $\mathbf{Y}$, formulated as $\mathbf{X}^s$ and $\mathbf{Y}^s$.

**Output:** aligned latent vectors $\mathbf{z}_x$ and $\mathbf{z}_y$, and an optimal transport plan $\mathbf{T}$.

uniPort performs the following steps:

1. Initialize coupled-VAE encoder $\psi$ and decoders $\phi, \phi_x, \phi_y$, and an OT plan $\mathbf{T} = \frac{1}{n_x n_y} \mathbf{1}_{n_x n_y}$ (optional).
2. For $m \leftarrow 1, \cdots, M$ do the following
   a. Randomly sample an integer index set $\mathcal{I} = (i_1, \cdots, i_{B_x}) \in [n_x]^{B_x}$ for dataset $\mathbf{X}$ and $\mathcal{J} = (j_1, \cdots, j_{B_y}) \in [n_y]^{B_y}$ for dataset $\mathbf{Y}$ without replacement, respectively.
   b. Initialize Minibatch-UOT plan $\mathbf{T}_B \in \mathbb{R}^{B_x \times B_y}$ by sampling rows and columns of $\mathbf{T}$, corresponding to $\mathcal{I}$ and $\mathcal{J}$, or uniform distribution when $\mathbf{T}$ is not specified.
   c. Input both $\mathbf{X}_{\mathcal{I}}^c$ and $\mathbf{Y}_{\mathcal{J}}^c$ through the shared probabilistic encoder $\psi$ to obtain $(\boldsymbol{\mu}_x, \boldsymbol{\sigma}_x)$ and $(\boldsymbol{\mu}_y, \boldsymbol{\sigma}_y)$, and then reparameterize $\mathbf{z}_x$ and $\mathbf{z}_y$ by $\mathbf{z}_x = \boldsymbol{\mu}_x + \boldsymbol{\sigma}_x \times v$ and $\mathbf{z}_y = \boldsymbol{\mu}_y + \boldsymbol{\sigma}_y \times v$;
   d. Reconstruct $\hat{\mathbf{X}}_{\mathcal{I}}^c$ and $\hat{\mathbf{Y}}_{\mathcal{J}}^c$ by decoder $\phi$, and $\hat{\mathbf{X}}_{\mathcal{I}}^s$ and $\hat{\mathbf{Y}}_{\mathcal{J}}^s$ by decoders $\phi_x$ and $\phi_y$, from $\mathbf{z}_x$ and $\mathbf{z}_y$.
   e. Compute minibatch transport cost $\mathbf{C}$ via Eq. (3) and obtain optimal Minibatch-UOT plan $\mathbf{T}_B^*$ via Eq. (4).
   f. Fix $\mathbf{T}_B^*$ and update $\psi, \phi, \phi_x$ and $\phi_y$ through back propagation via minimizing $\mathcal{L}_{\text{UOT}^*}$ and $-\mathcal{L}_{\text{ELBO}^*}$.
   g. Update $\mathbf{T}$ with rows and columns as $\mathbf{T}_B^*$, corresponding to $\mathcal{I}$ and $\mathcal{J}$ (optional).

## Training details

uniPort consists of one encoder and three decoders for the integration of two datasets. The encoder is a two-layer neural network (1024–128) with the ReLU activation function. The decoders have no hidden layer, but directly connect the 16-dimensional latent variables to the output layers with the *Sigmoid* activation function. The *Adam* optimizer with a 5e-4 weight decay is used to maximize the ELBO. Minibatch size is 256. We set all the training with learning rate to 2e-4 and optimal transport entropy regularization parameter $\epsilon$ to 0.1. We chose parameters for training coupled-VAE and Minibatch-UOT from $\lambda \in \{0.5, 1.0, 5.0\}$, $\gamma \in \{0.5, 1.0\}$, and $\lambda_s \in \{0.5, 1.0\}$ for all datasets. uniPort is robust to different choices of $\lambda, \lambda_s$ and $\gamma$ (Supplementary Fig. 17a). It is also robust to the number of selected common and dataset-specific highly variable genes (HVGs) within a certain range (Supplementary Discussion 6 and Supplementary Fig. 17b). The default maximum number of training iterations is 30,000, and an early stopping is triggered when no improvement has occurred for 30 epochs. Our experimental environment includes two AMD EPYC 7302 16-Core Processors, 128GB DDR4 memory, and a Tesla T4 NVIDIA GPU with 16GB memory.

## Data preprocessing

- **Human PBMC multi-omics dataset**. The paired multi-omics PBMC dataset measuring both DNA accessibility and gene expression data were downloaded from the publicly available 10× Genomic datasets[36]. The raw gene expression data were processed using the Seurat package (v4.1.0)[9]. Cells and genes were filtered under default parameters. SCTransform was used for normalization. For paired scATAC-seq data, the fragment files were qualified using Signac (v1.5.0)[58] under default parameters. Peaks were called with MACS2[61]. Then latent semantic indexing (LSI) was used for dimensionality reduction, resulting in the binary cell-by-bin accessibility as input for TF-IDF weighting. Consequent dimensionality reduction used Singular Value Decomposition (SVD). The gene activity count matrices were derived using MAESTRO[37]. Cell types were arranged by transferring labels from an annotated PBMC reference dataset[6] using Seurat. Finally, 11,259 cells with 28,307 features in scATAC and 11,942 genes in scRNA were used for the integrative analysis.

- **microfluidic-based PBMC dataset**. The unpaired microfluidic-based PBMC dataset was obtained from MAESTRO[37], including 1919 cells with 28307 genes in gene activity score of scATAC and 1985 cells with 1477 genes in scRNA. The gene activity score was processed by MAESTRO.

- **Mouse spleen dataset**. The processed and annotated scRNA data of mouse spleen was directly obtained from MultiMAP[13], containing 4382 cells with 13,575 genes. The gene activity score of scATAC[62] was processed by SnapATAC[63], containing 3166 cells across 19,410 genes.

- **Synthetic STARmap dataset**. The synthetic STARmap dataset was obtained from the benchmarking paper[29], containing 14249 cells with 34041 genes in scRNA data and 189 cells with 882 genes in the spot data.

- **Brain scRNA and MERFISH dataset**. The scRNA-seq (10×) dataset from the preoptic region of the hypothalamus in six mice was obtained from NCBI GEO accession number GSE113576[64]. The MERFISH data and annotations were downloaded from Dryad repositories [https://datadryad.org/stash/dataset/doi:10.5061/dryad.8t8s248]. Naive female mouse #1 in the MERFISH data was chosen for integration, and the mouse #2 was extracted to validate the result of online imputation. All cells labeled as 'Ambiguous' or 'Unstable' were removed from both datasets. scRNA data were qualified and processed using Seurat as above.

- **10× Visium dataset and reference**. The series of 10× Visium spatial datasets, including sagittal mouse brain slice (2696 spots with 48,721 genes) and human breast cancer (2518 spots with 17,943 genes), were obtained from 10× Genomic datasets[36]. Spaceranger outputs were obtained and processed with Seurat to generate the standard gene expression matrix. For the mouse brain dataset, the annotated scRNA dataset from adult mouse cortical cell taxonomy from the Allen Institute was chosen as reference, and the subset used was extracted from SPOTlight tutorial[43]. The processed and labeled scRNA reference for breast cancer slice came from[65].

- **Pancreatic ductal adenocarcinoma dataset**. The microarray-based spatial transcriptomic dataset of pancreatic ductal adenocarcinoma (PDAC) was acquired from NCBI GEO accession number GSE111672[51]. The spatial dataset of batch A was adopted together with scRNA-seq generated from the same sample. Cell types in the scRNA reference had already been assigned. These two expression profiles were set as input for uniPort.

- **uniPort preprocessing steps**. uniPort preprocessed data as follows: (1) We filtered out cells with fewer than 200 genes and filtered out genes observed in fewer than 3 cells for PBMC data. No cells or genes were filtered out for other data. (2) We normalized total counts of each cell using the *scanpy.pp.normalize_total* function in the scanpy package[66] in Python. (3) We performed log-normalization of all datasets with an offset of 1 using the *scanpy.pp.log1p* function. (4) We identified $k = 2000$ highly variable common genes across cells of all datasets, and identified $k_x = k_y = 2000$ highly variable genes for each dataset using *sc.pp.highly_variable_genes* function, respectively. (5) We normalized values of each gene to the range of 0-1 within each dataset by the *MaxAbsScaler* function in the *scikit-learn* package in Python. The processed matrix was used as input for uniPort.

## Settings used in comparing methods

For the integration among scATAC-seq, scRNA-seq and spatial transcriptomic datasets, we benchmarked the performance of uniPort (v1.1.1) against the following methods: Seurat (v4.1), Liger (v1.0.0), and Harmony (v1.0) in R environment (v4.0.2), SCALEX (v0.2.0), MultiMAP (v0.0.1), scVI (v0.17.1), scMC (v1.0.0), GLUE (v0.2.3), SCOT (v1.0), Pamona (v0.1.0), and Tangram (v1.0.3) in Python environment (v3.8.13). We compared their co-embedding UMAP visualization with parameters *n_neighbors=15* and *min_dist=0.1*. All methods adopted gene activity matrices derived from the above data preprocessing steps as input for scATAC-seq. Detailed parameters used in each method are as follows.

- **Seurat**. The Seurat package (v4.1.0) was used for all datasets. The raw gene expression profile and unnormalized gene activity matrix were set as input. The matched matrices were log-normalized using the *NormalizeData* function in Seurat with scaling at 10000 for cell-level normalization separately. Then the *FindVariableFeatures* function was used to pick the top 2000 HVGs for scRNA. The anchors between scRNA and scATAC were acquired using the *FindTransferAnchors* function, where 'cca' was set as the reduction method with features from HVGs in the scRNA dataset. The imputed scATAC data used the *TransferData* function, where anchors were weighted by latent semantic indexing. Then scRNA and imputed scATAC datasets were merged and reduced using PCA with the *RunPCA* function under default 50 principal components (PCs). UMAP was adopted for visualization at 30 PCs using the *RunUMAP* function. For MERFISH data, the *FindIntegrationAnchors* and *IntegrateData* functions were used for integration after common feature selection with the *SelectIntegrationFeatures* function. Then the integrated matrix was scaled using the *ScaleData* function, and PCA and UMAP were applied to all at 30 PCs. For the 10X Visium datasets, the cell-type assigning probabilities of spots were derived using the *FindTransferAnchors* and *TransferData* functions, where spatial datasets were reduced using PCA at 30 PCs.

- **Liger**. The R package rliger (v1.0.0) was used for integrating scATAC and MERFISH data with scRNA data. The normalization and selection of HVGs are the same as above steps in Seurat. For all datasets, the number of factors was set to 20 at default in the *optimizeALS* function, which was done by calling the *RunOptimizeALS* function in the SeuratWrappers package (v0.3), and then building a shared factor neighborhood graph and quantile normalization of corresponding clusters applied through the *RunQuantileNorm* function.

- **Harmony**. The R package harmony (v1.0) was used for integrating scATAC and MERFISH data with scRNA data. The normalization and selection of HVGs are the same as above steps in Seurat. Then PCA was performed using the *RunPCA* function in Seurat at 50 PCs. Harmony refined the results of PCA at top 30 PCs.

- **SCALEX**. The Python package scalex (v0.2.0) was used for integrating scATAC and MERFISH data with scRNA data. The data preprocessing steps are the same in uniPort and SCALEX. Therefore, we input the same preprocessed data without common genes as those in uniPort. Embedded preprocessing of the scalex package was ignored. We ran SCALEX using the *SCALEX* function with parameters set as default.

- **MultiMAP**. The Python package MultiMAP (v0.0.1) was used for integrating scATAC and MERFISH data with scRNA data. We followed the tutorials of MultiMAP on GitHub. We first applied the *sc.pp.normalize_total* and *sc.pp.log1p* functions to raw counts without scaling. We then processed data with the *MultiMAP.TFIDF_LSI* function for dimensionality reduction of ATAC peaks, *sc.pp.scale* and *sc.pp.pca* functions in Python package scanpy for dimensionality reduction of scRNA and MERFISH data, and the *MultiMAP.Integration* function for integration with all parameters set as default, as suggested in the pipeline in the GitHub repository.

- **scVI**. The Python package scvi-tools (v0.17.1) was used for integrating scATAC and MERFISH data with scRNA-seq data. We followed the tutorials of scVI on GitHub. We first performed *sc.pp.normalize_total* and *sc.pp.log1p* functions for input data. Then we selected the top 2000 highly variable genes using the *sc.pp.highly_variable_genes* functions with *flavor="seurat_v3"*. After that, *scvi.model.SCVI* and *scvi.model.SCVI.setup_anndata* functions were used for model initialization.

- **scMC**. The R package scMC (v1.0.0) was used for integrating scATAC-seq and scRNA-seq data. The normalization and selection of HVGs are the same as the above steps in Seurat. Then we scaled data by *ScaleData* function and performed scMC through *RunscMC* function. We searched parameter *similarity.cutoff* of *RunscMC* function within {0.1, 0.2, 0.3, 0.4, 0.5, 0.6, 0.7, 0.8, 0.9} for the best integration result. All other parameters were set as default.

- **GLUE**. The Python package scglue (v0.2.3) was used for integrating PBMC and mouse spleen examples. We followed the tutorials of GLUE on GitHub. We first performed *sc.pp.normalize_total*, *sc.pp.log1p* and *sc.pp.scale* functions for input data with default parameters for scRNA data. Then we used *sc.tl.pca* function with *n_comps=100* and *scglue.data.lsi* function with *n_components=100* for scRNA and scATAC data, respectively. We chose *prob_model='ZINB'* for the PBMC example and *prob_model='Normal'* for mouse spleen example in *scglue.models.configure_dataset* function. All other parameters were set as default.

- **SCOT**. We downloaded SCOT (v1.0) from https://github.com/rsinghlab/SCOT for integrating scATAC-seq with scRNA-seq data. Because SCOT did not provide a data preprocessing tutorial, we compared different approaches and used the one with the best result. To be specific, we performed *uniport.TFIDF_LSI* function with default parameters for ATAC peaks, which reduced the dimension to 49. Then we performed *sc.pp.normalize_total*, *sc.pp.log1p* and *sc.pp.pca* functions with default parameters for RNA data. We set all parameters as default in *scot.align* function.

- **Pamona**. The Python package pamona (v0.1.0) was used for integrating scATAC-seq and scRNA-seq data. All data preprocessing was performed in the same manner as SCOT.

- **Tangram**. The Python package tangram-sc (v1.0.3) was used for imputing MERFISH through scRNA-seq data. We followed the tutorial of Tangram on GitHub. First, *tg.pp_adatas* function was applied to pre-process both training MERFISH and scRNA data. Then, we performed *tg.project_genes* function to find the project matrix, which was used to impute MERFISH test genes. All parameters were set as default.

- **gimVI**. The model gimVI in the Python package scvi-tools (v0.17.1) was used for imputing MERFISH through scRNA-seq data. We followed the tutorial of scvi-tools in GitHub, and used *GIMVI.train* function to train the model in 200 epochs. All parameters were set as default.

## Evaluation metrics

The accuracy of cell-type assigning was quantified by the adjusted Rand Index (ARI), Normalized Mutual Information (NMI) and F1 score.

To be specific, we trained a *k*-Nearest-Neighbor (*k*NN) classifier by the *sklearn.neighbors.KNeighborsClassifier* function based on cell-type annotations and UMAP coordinates of reference data (e.g., scRNA) embeddings in the common latent space. Then, we applied the well-trained *k*NN classifier to predict cell-type annotations of query dataset (e.g., scATAC/MERFISH) embeddings, and calculated the ARI, NMI and F1 scores by real and predicted query cell type annotations. All results are based on UMAP visualization with parameters *n_neighbors=15* and *min_dist=0.1*.

- **Adjusted Rand Index**. Adjusted Rand Index (ARI) is introduced to determine whether real and predicted cell-type clusters are like each other. The Rand Index (RI) computes the similarity by taking all points identified within the same cell-type cluster. The Adjusted RI (ARI) is the chance-corrected version of the Rand index and calculated with RI as

$$ARI = \frac{RI - \text{expected RI}}{\max(RI) - \text{expected RI}}.$$

  The ARI value ranges from 0 to 1, with 0 for random labeling and 1 for perfect matching.

- **Normalized Mutual Information**. Normalized Mutual Information (NMI) is a variant of a common measure in information theory called Mutual Information. It is calculated as

$$NMI(U, V) = \frac{2 \times I(U; V)}{H(U), H(V)},$$

  where U and V are categorical distribution for the real and predicted cell-type annotations, I is the mutual entropy function and H is the Shannon entropy function.

- **F1**. The F1 score combines the precision and recall of a classifier into a single metric by taking their harmonic mean. It is calculated as

$$F1 = \frac{2 \times P*R}{P + R},$$

  where P is the precision and R is the recall of the *k*-NN classifier.

To assess the separation of clusters and batch mixing, the Silhouette coefficient and Batch Entropy score were adopted. Furthermore, we employed average FOSCTTM[15] to assess cell-to-cell neighborhood preservation for paired datasets profiled from the same cells. All these systematic benchmarks were applied in scRNA-seq, scATAC-seq and MERFISH datasets. All results are based on UMAP visualization with parameters *n_neighbors=15* and *min_dist=0.1*.

- **Silhouette coefficient**. The Silhouette coefficient is calculated using the mean intra-cluster distance (a) and the mean nearest-cluster distance (b) for each sample. The Silhouette coefficient for a sample is calculated as

$$Silhouette = \frac{b - a}{\max(a, b)}.$$

  By default, we scale the score between 0 and 1 by

$$Silhouette \leftarrow (Silhouette + 1)/2,$$

- **Batch Entropy score**. It was derived from SCALAX[25] inspired by "entropy of batch mixing"[33]. It evaluates the sum of regional mixing entropies at the location of randomly chosen cells

from different datasets where a high score indicates cells from various datasets mixing well. This can be calculated as

$$p_i' = \frac{p_i/P_i}{\sum_{i=1}^{n} p_i/P_i},$$

$$E = \sum_{i=1}^{n} p_i' \log(p_i'),$$

where $P_i$ is the proportion of cell numbers in each batch to the total cell numbers, and $p_i$ is the proportion of cells from batch *i* in a given region. We calculated the Batch Entropy score only based on cells from cell types that are common in different batches.

- **average FOSCTTM**. FOSCTTM refers to "fraction of samples closer than the true match". It was used to evaluate the relationship preservation of cell-to-cell pairings' neighborhood. It was calculated on two datasets with known cell-to-cell correspondence information. The average FOSCTTM is calculated as

$$\text{average FOSCTTM} = \frac{1}{2n} \text{FOSCTTM},$$

$$FOSCTTM = \sum_{i=1}^{n} \frac{s^i}{n} + \sum_{i=1}^{n} \frac{t^i}{n},$$

$$s^i = |\{j | d(z_{x_j}, z_{y_i}) < d(z_{x_i}, z_{y_i})\}|$$

$$t^i = |\{j | d(z_{x_i}, z_{y_j}) < d(z_{x_i}, z_{y_i})\}|$$

where *n* is the number of both dataset **X** and **Y**, $z_{x_i}$ and $z_{y_i}$ are paired cells, $s^i$ and $t^i$ are the number of cells closer to the *i*-th cell in a dataset than its true match in another dataset. The average FOSCTTM ranges from 0 to 1, and lower values indicate higher accuracy.

**Statistics & reproducibility**

All statistical calculations were implemented in R (v4.0.2; https://cran.r-project.org/). The detailed statistical tests were indicated in figures or associated legends where applicable. No statistical method was used to predetermine sample size. No data were excluded from the analyses. Complete randomization was performed for allocating groups. Our study does not involve group allocation that requires blinding.

**Reporting summary**

Further information on research design is available in the Nature Portfolio Reporting Summary linked to this article.

## Data availability

All data analyzed in this article are publicly available through online sources. We present links to all data sources in Supplementary Data 1. Source data are provided as a Source Data file. Source data are provided with this paper.

## Code availability

The uniPort framework was implemented in the 'uniport' Python package, which can be installed through PyPI [https://pypi.org/

project/uniport/], and its open-source code is maintained at https://github.com/caokai1073/uniPort.

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

## Acknowledgements

This work was supported by the National Key Research and Development Program of China (No. 2019YFA0709501 to L.W.), the National Natural Science Foundation of China (Nos. 61733018, 62173250 to Y.H., and No. 12071466 to L.W.), Shanghai Municipal Science and Technology Major Project (No. 2021SHZDZX0100 to Y.H.), and the Fundamental Research Funds for the Central Universities (No. 22120200046 to Y.H.).

## Author contributions

Y.H. and L.W. conceived and supervised the project. K.C. conceived, designed and implemented the uniPort model. Q.G. validated and analyzed the results. K.C., Q.G. and L.W. wrote the paper.

## Competing interests

The authors declare no competing interests.
