## [Peer Review File · Nature Communications]

nature portfolio

Peer Review File

A unified computational framework for single-cell data integration with optimal transportREVIEWER COMMENTS

Reviewer #1 (Remarks to the Author):

The manuscript proposes a new method uniPort for integrating unpaired single-cell multiomics data based on variational autoencoder (VAE). The architecture of uniPort contains separate decoders for genes common to the datasets as well as dataset-specific genes, and achieves alignment in the latent space through optimal transport (OT). I find the method contains a few interesting points. However, given the rapidly increasing collection of published methods on single-cell multiomics integration, I am wondering if the authors can further justify the novelty and unique contributions of uniPort.

- Comparison with recent integration method (VAE based) for unpaired data. [1] below is an example, where the VAE achieves both integration and regulatory inference, and does not require conversion of ATAC-seq features into gene activity scores. Discussions about conceptual differences and ideally benchmarking on the PBMC example will be helpful.

- Different versions of OT are commonly used in for single-cell integration and trajectory inference. Is the OT used in uniPort different from the existing literature? If not, giving references would help with providing a context for the current method.

- Could the authors comment on the use of gene activity scores and whether it results in any information loss for determining subtle cell states.

- For the PBMC example, commonly used metrics such as FOSCTTM should be used to assess whether two modalities from the same cell have been paired properly.

- Computing the correlation between the predicted and real mean gene expression in Fig 3c is not particularly informative. It would be more helpful to compare the correlations of different marker genes between predicted and real. How sensitive are the results to the number of common and data-specific HVG selected? What happens if the query data contains cell types not contained in the reference?

- In terms of the applications, I find the spatial transcriptomics ones more interesting. However, at the same time, deconvolution of spatial data is again a field with a plethora of methods, some of which require matching scRNA-seq data while others do not. Does uniPort provide novel functionalities lacking in the other methods? Discussions and benchmarking would be helpful here.

1. Cao, Zhi-Jie, and Ge Gao. "Multi-omics single-cell data integration and regulatory inference with graph-linked embedding." *Nature Biotechnology* (2022)

Reviewer #2 (Remarks to the Author):

In this work, Cao et al. present uniPort, a new framework to integrate single-cell multi-omics data. The framework combines a coupled variational autoencoder (coupled-VAE) and a mini-batch unbalanced optimal transport (minibatch-UOT) plan, which are added together to define the total loss function to optimize. The authors compare uniPort to several established frameworks for single-cell integration, such as Seurat, LIGER, MultiMap, and scVI, to a range of useful case studies. In particular, the authors demonstrate uniPort's advantages in cell type classification when integrating single-cell transcriptomics, chromatin accessibility and spatially resolved transcriptomic data

The combination of a variational autoencoder and optimal transport—two frameworks that are becoming increasingly prominent in single-cell analysis—is interesting and the methods and results are well presented. Furthermore, the choice of datasets for comparison is entirely appropriate and varied. However, I think the reasoning for why combining the VAE and OT frameworks is useful need to be expanded upon significantly. In addition, the paper needs to consider and compare more recent multi-omic integration methods, as UOT has been used for integration of multi-omics data.

There are several comments I wish the authors to address:

Major comments:

- Comparing uniPort to a wider range of multi-omics-based methods is critical. In particular, it would be worth considering: more recent methods in Seurat ([https://www.cell.com/cell/fulltext/S0092-8674\(21\)00583-3](https://www.cell.com/cell/fulltext/S0092-8674(21)00583-3)); scMC (<https://doi.org/10.1186/s13059-020-02238-2>); while the scVI-tools framework has a tool, totalVI, that is specifically for multi-omic integration (<https://doi.org/10.1038/s41592-020-01050-x>).
- It would be worthwhile to compare uniPort to other OT-based methods, such as SpaOTsc (<https://doi.org/10.1038/s41467-020-15968-5>) and SCOT (<http://doi.org/10.1089/cmb.2021.0446>), which are used for spatial deconvolution and multi-omics integration, respectively, as uniPort is the first method that I know of that uses mini-batch UOT for single-cell analysis.
- The authors state that minibatch OT is used to allow uniPort to be applied to larger datasets, which are common in single-cell analysis. However, I do not see any comparison of runtime between the different considered integration frameworks. I think that would help strengthen the methods benchmarking.
- One of the proposed features of uniPort is online prediction after constructing a reference atlas, which is demonstrated by predicting scRNA-seq gene expression onto spatial MERFISH data, which has significantly lower gene coverage. I think comparing uniPort to other methods specialized for mapping scRNA-seq gene expression onto spatial data, such as Tangram or gimVI, which were considered in a recent benchmarking study (<https://doi.org/10.1038/s41592-022-01480-9>).
- For Minibatch-UOT, the parameter ρ related to marginal penalization in equation (4) plays an important role on the integration, especially when there are rare cell types that are present in only one dataset. What is the reason to set $\rho=1$? When the value of ρ is changed, can uniPort still separate Ependymal cells, which are only observed in MERFISH data? Do the results of Figure 3a change significantly?
- The input and output of uniport is unclear, e.g. XC , XS... in Figure 1, as the notation for input and output is the same. Similarly, in the Methods section, uniPort also requires XS, YS as input, suggesting that uniPort does not generate data-specific genes but rather requires them as input.
- To predict genes in scRNA-seq data by the trained reference atlas (Line 197), only 153 common genes are used as input. How do the authors consider and deal with possible batch effects between samples?
- When using evaluation metrics for benchmarking, I think it would help that the authors were clearer about the comparison. For example, are the authors evaluating known clustering vs predicted clustering? For the single-cell RNA-sequencing and single-cell ATAC-sequencing data of peripheral mononuclear blood cells, do the authors consider the consistency of paired single cell labels?

Minor comments:

- **Please double check the figure captions. For example, in Figure 3a, only the integration results from uniPort are shown, rather than the results from Harmony, Seurat, SCALEX or scVI.**
- **It is good to see the comparison between gene expressions in MERFISH and predicted RNA as in Supplementary Fig 7. What about the real single cell RNA expression? It seems like the authors only show the comparison the mean expressions between real and predicted RNA.**
- **In the Supplementary information, line 55: 'bisaed', which I think should be 'biased'. I think this is the only spelling error, but please do check over the rest of the manuscript to make sure.**

Responses to Reviewers:

We would like to thank the reviewers for their thoughtful comments and valuable suggestions on our manuscript. The manuscript has been fully revised by following the reviewers' comments and suggestions, making our results more solid and convincing.

The point-by-point responses to the reviewers' comments and suggestions are as follows and our responses are in red color.

Reviewer #1:

Overall comments

R1: The manuscript proposes a new method uniPort for integrating unpaired single-cell multiomics data based on variational autoencoder (VAE). The architecture of uniPort contains separate decoders for genes common to the datasets as well as dataset-specific genes, and achieves alignment in the latent space through optimal transport (OT). I find the method contains a few interesting points. However, given the rapidly increasing collection of published methods on single-cell multiomics integration, I am wondering if the authors can further justify the novelty and unique contributions of uniPort.

Response:

We sincerely appreciate the highly insightful and constructive suggestions by Reviewer #1 so that we can improve the manuscript accordingly.

In the revised manuscript, we addressed the novelty and unique contributions of our

in the revised manuscript, we addressed the novelty and unique contributions of our work in both methods and applications.

- In terms of methods, we have innovatively introduced minibatch unbalanced optimal transport into VAE to overcome the limitation on conventional VAE for single-cell heterogeneous and/or unpaired data integration. To the best of our knowledge, uniPort is the first method to use minibatch unbalanced optimal transport in single-cell genomics analysis.
- In terms of applications, our uniPort is an accurate, robust and efficient computational platform that can not only accurately integrate heterogeneous single-cell datasets, but also impute genes to enhance the resolution of spatial data as a generative model, and deconvolute barcoding-based spatial data through an optimal transport plan.

We provided the detailed one-to-one responses below.

Major comments:

R1.1: Comparison with recent integration method (VAE based) for unpaired data. [1] below is an example, where the VAE achieves both integration and regulatory inference, and does not require conversion of ATAC-seq features into gene activity scores. Discussions about conceptual differences and ideally benchmarking on the PBMC example will be helpful.

Response: We thank the reviewer for the valuable advice. There are several conceptual differences between uniPort and GLUE:

1. uniPort used the gene activity score of scATAC by MAESTRO or Signac (lines 337-348 on page 13), while GLUE developed a sophisticated knowledge-based graph that models regulatory signals of scATAC instead of using the gene activity score.
2. GLUE used a discriminator to align the latent distributions of different modalities with a complex mechanism introduced for unbalanced matching. In comparison, uniPort employed an efficient unbalanced optimal transport to align the distributions.
3. GLUE is specifically designed for multi-omics integration with a prior gene regulatory relationship, while uniPort can not only integrate multi-omics data, but also impute marker genes for one modality by another as well as deconvolute barcoding-based spatial transcriptomics data.

Benchmark:

In the revised version, we further benchmarked uniPort against GLUE on scATAC and scRNA from two datasets of PBMC and one dataset of mouse spleen (Figs. 2, 3, 4 and Supplementary Figs. 2, 3). uniPort achieved comparable results with GLUE on the two PBMC datasets and outperformed GLUE on the mouse spleen dataset, essentially in the case of integrating heterogeneous datasets. For more details about the comparison, please see lines 120-143 and lines 157-165 on pages 5-6. For details on GLUE implementation, please see lines 567-573 on page 21.

R1.2: Different versions of OT are commonly used in for single-cell integration and trajectory inference. Is the OT used in uniPort different from the existing literature? If not, giving references would help with providing a context for the current method.

Response: We thank the reviewer for this insightful comment. In the revised manuscript, we extensively discussed and compared uniPort with existing OT-based methods. Our method is significantly different from the existing OT methods for single-cell analysis. To the best of our knowledge, prevalent OT-based methods for single-cell analysis are based on global optimal transport. In this manuscript, we first introduced a minibatch unbalanced OT and combined it with the VAE model for this field, which reduces the computational cost of both time and memory of OT. We emphasized the differences in the revised manuscript through the following points.

- We benchmarked uniPort against SCOT and SpaOTsc. SCOT and SpaOTsc are two state-of-the-art OT-based methods for single-cell multi-omics integration and spot deconvolution, respectively. We compared uniPort with SCOT on the integration of two datasets of PBMC and one dataset of mouse spleen. We found uniPort outperformed SCOT in data integration on all examples (lines 120-143 on pages 5-6) while consuming much less computational time than

SCOT (Supplementary Fig. 11). We also compared uniPort with SpaOTsc for deconvolution of the synthetic STARmap data, and we found that uniPort achieved higher accuracy and is more robust in performance using the evaluation metrics (Supplementary Fig. 7; lines 225-233 on page 9).

- We provided a new paragraph for OT methods comparison.
In the revised manuscript, we added a new paragraph (lines 327-336 on page 12): “To the best of our knowledge, prevalent OT-based methods for single-cell analysis are based on global optimal transport, e.g., SCOT and Pamona for single-cell multi-omics data integration, SpaOTsc and novoSpaRc for spatial positions reconstruction, and Waddington-OT for trajectory inference. Global optimal transport makes the computation very expensive. To resolve this drawback, our uniPort firstly introduces a Minibatch-UOT into a VAE-based framework for single-cell genomics analysis, which only needs to solve a mini-batch transport plan at each iteration, thus significantly reducing the computational cost. Therefore, it is scalable to large datasets (Supplementary Fig. 11, 12). Additionally, in general, our coupled-VAE and Minibatch-UOT-based uniPort is more accurate than other OT-based methods, such as SCOT, Pamona and SpaOTsc in different tasks.”

R1.3: Could the authors comment on the use of gene activity scores and whether it results in any information loss for determining subtle cell states.

Response: We thank the reviewer for pointing out the issue of modeling gene activity scores.

In the revised version, we addressed this issue of the importance of modeling gene activity scores through a new task: we tested uniPort’s performance when the gene

activity score was calculated by different models of MAESTRO and Signac, respectively. The integration result showed that uniPort achieved better performance on gene activity score by MAESTRO with all evaluated metrics higher than that by Signac (Supplementary Fig. 13), which demonstrated the importance of modeling gene activity score.

In the revised manuscript, we commented on the use of gene activity scores as follows (lines 337-348 on page 13):

“Given that our integration of scATAC is based on gene activity score, we also tested uniPort's performance when gene activity score is calculated in different approaches. We employed two methods to form the gene activity score introduced by Signac and MAESTRO, respectively. Signac defines gene activity score for scATAC as the read count in the gene body and promoter region. MAESTRO calculates gene activity score as a weighted sum of nearby cis-regulatory elements (REs), where the weight is an exponentially decreasing function of distances of REs and target genes. The integration result showed that uniPort achieved better performance on the gene activity score by MAESTRO with all evaluated scores higher than that by Signac (Supplementary Fig. 13), which demonstrated the importance of modeling the gene activity score. It is worth noting that, GLUE, which is developed based on a sophisticated knowledge-based graph that explicitly and accurately models regulatory signals of scATAC, instead of using gene activity score, provides an important technique for analyzing scATAC data.”

Supplementary Figure 11. Comparison of uniPort performance over different gene activity scores.

R1.4: For the PBMC example, commonly used metrics such as FOSCTTM should be used to assess whether two modalities from the same cell have been paired properly.

Response: We thank the reviewer for the suggestion of the FOSCTTM metric for the evaluation of paired data integration. In the revised version, we employed the average FOSCTTM for performance evaluation as follows.

- We added the description of the average FOSCTTM in the main text (lines 624-630 on page 24)
- We evaluated the average FOSCTTM on the paired PBMC example in Fig. 2e. uniPort achieved an average FOSCTTM of 0.0694, ranked second among 11 compared methods, slightly below GLUE (0.0441) while significantly higher than other methods (lines 124-126 on page 5).

Fig. 2: e, Comparison of average FOSCTTM of different methods.

R1.5: Computing the correlation between the predicted and real mean gene expression in Fig 3c is not particularly informative. It would be more helpful to compare the correlations of different marker genes between predicted and real.

Response: We thank the reviewer for the suggestion. To better test the accuracy of gene prediction by uniPort, we made modifications by adding the following revisions.

1. We predicted RNA genes by MERFISH data for mouse 2, and as suggested, we further calculated the Pearson correlation coefficient (PCC) between MERFISH marker genes and the predicted RNA marker genes (Supplementary Fig. 6c). The

result demonstrated high PCC between the same genes.

C

Supplementary Fig. 6: c, Pearson correlation coefficient between imputed marker genes and real MERFISH marker genes of mouse 2.

2. We further conducted a new task as another reviewer suggested (lines 183-212 on pages 7-8). We followed the scheme of gimVI to impute missing genes in MERFISH from scRNA. To be specific, we first randomly selected 80% (i.e., 122/153) genes in MERFISH as training genes and reserved the remaining 20% (i.e., 31/153) genes as testing genes. We repeated the above steps twelve times and obtained 12 training and testing gene sets. Afterwards, we trained the uniPort network with each training gene set, and then imputed the corresponding testing

gene set. We compared our results with two state-of-the-art gene-imputing methods: gimVI and Tangram. We applied uniPort, gimVI and Tangram to impute testing genes, and used UMAP to visualize both training and testing genes (Fig. 5e). With an imputation framework like that of gimVI, we also expected imputed values of uniPort to carry gene-specific biases from scRNA genes. Therefore, for performance evaluation, we followed gimVI and reported the median and average Spearman correlation coefficients (mSCC and aSCC), as well as the median and average Pearson correlation coefficients (mPCC and aPCC) over imputed and ground truth testing genes. uniPort provided a significant improvement over the two compared methods on the MERFISH dataset. For example, uniPort separated different cell types in the UMAP visualization of imputed genes with a better result (Fig. 5e), and demonstrated the highest mSCC (0.259), aSCC (0.26), mPCC (0.249) and aPCC (0.294) (Fig. 5f), significantly above those of gimVI (mSCC of 0.221, aSCC of 0.24, mPCC of 0.201 and aPCC of 0.242) and Tangram (mSCC of 0.188, aSCC of 0.206, mPCC of 0.202 and aPCC of 0.231).

Fig. 5: **e**, UMAP visualization of imputed MERFISH genes of Tangram, gimVI and uniPort.; **f**, Median and average Spearman correlation coefficient (SCC) and Pearson correlation coefficient (PCC) between real and imputed MERFISH genes.

R1.6: How sensitive are the results to the number of common and data-specific HVG selected?

Response: We thank the reviewer for the valuable suggestion. In the revised version, we evaluated the robustness of uniPort on the number of common and data-specific HVG selected on the paired PBMC example with three scenarios as follows (Supplementary Fig. 17b):

- (1) We selected 8000, 4000, 2000, 1000, and 500 genes, respectively, for both common and dataset-specific HVGs for integration;
- (2) We selected 8000, 4000, 2000, 1000, and 500 genes, respectively, for dataset-specific HVGs, while maintaining the number of common HVGs as 2000 for integration;
- (3) We selected 8000, 4000, 2000, 1000, and 500 genes, respectively, for common HVGs, while maintaining the number of common HVGs as 2000 for integration.

The result shows that uniPort is robust to different choices of the number of common and data-specific HVGs (lines 449-451 on page 17; Supplementary Result 6).

Supplementary Fig. 17: **b**, evaluated scores of uniPort when selecting different numbers of common and specific HVG over the paired PBMC example.

R1.7: What happens if the query data contains cell types not contained in the reference?

Response: We thank the reviewer for pointing out this issue.

uniPort maps query data to reference data in the latent space by minimizing minibatch unbalanced optimal transport loss. Therefore, if a cell type in the query is unique, the cells of this type can be identified as unbalanced parts and have low transport mass to any cell in the reference dataset. In the revised manuscript, we evaluated uniPort on this scenario with two supporting experimental results:

1. In the MERFISH example, ependymal cells are unique in the MERFISH data, while we set the scRNA data as a reference. uniPort accurately separated ependymal cells from other cell types (Fig. 5a, b; see lines 176-178 on page 7).

Fig. 5: **a**, UMAP visualization of MERFISH and scRNA data before integration; **b**, UMAP visualization of MERFISH and scRNA data after uniPort integration.

2. We additionally evaluated uniPort on two unbalanced matching tasks on the mouse spleen dataset (lines 147-165 on page 6): To evaluate the performance of uniPort on heterogeneous data integration, we conducted two unbalanced matching tasks by removing some cell types from scATAC or scRNA of mouse spleen, separately. We found that uniPort is more robust than the other methods when heterogeneity is presented in the datasets.

Figure 4: **uniPort integrates cell-type unbalanced mouse spleen.** **a**, UMAP visualization of the case of “UBM-ATAC” after uniPort integration colored by omics and cell annotations. **b**, UMAP visualization of the case of “UBM-RNA” after uniPort integration. **c**, Comparison of total scores of ARI, NMI and F1 of different methods in three cases. **d**, Comparison of Batch Entropy scores and Silhouette coefficients of different methods in three cases.

R1.8: In terms of the applications, I find the spatial transcriptomics ones more interesting. However, at the same time, deconvolution of spatial data is again a field with a plethora of methods, some of which require matching scRNA-seq data while others do not. Does uniPort provide novel functionalities lacking in the other methods? Discussions and benchmarking would be helpful here.

Response: We appreciate the reviewer’s comment about our applications in the

responses to appreciate the reviewer's comments about our applications in the deconvolution of spatial data.

uniPort does provide novel functionalities lacking in the other method. For example, online data integration ability is becoming increasingly important for single-cell experiments. uniPort is a generative model based on the VAE model, which enables it to integrate single-cell data and impute genes in an online manner. We exemplified this function of uniport on lines 208-212 of page 8. In contrast, to the best of our knowledge, prevalent deconvolution methods (e.g., those which were benchmarked in a recent benchmarking study ([29] in our reference)) cannot perform such a task in an online manner. We will further develop new functionalities which can be facilitated by the date generative framework of uniPort.

Meanwhile, in the revised manuscript, we benchmarked uniPort on a synthetic STARmap against two state-of-the-art spatial deconvolution methods: Tangram and SpaOTsc, as another reviewer suggested (lines 213-233 on pages 8-9). uniPort achieved better performance than another OT-based method SpaOTsc.

Supplementary Figure 7. Spatial deconvolution benchmark. Pearson correlation coefficient (PCC), structural similarity index (SSIM), root mean square error (RMSE) and Jensen-Shannon divergence (JSD) of uniPort, Tangram and SpaOTsc over simulated STARmap example. Higher PCC and SSIM, and lower RMSE and JSD, indicate better performance.

Reviewer #2

Overall comments

R2: In this work, Cao et al. present uniPort, a new framework to integrate single-cell multi-omics data. The framework combines a coupled variational autoencoder (coupled-VAE) and a mini-batch unbalanced optimal transport (minibatch-UOT) plan, which are added together to define the total loss function to optimize. The authors compare uniPort to several established frameworks for single-cell integration, such as Seurat, LIGER, MultiMap, and scVI, to a range of useful case studies. In particular, the authors demonstrate uniPort's advantages in cell type classification when integrating single-cell transcriptomics, chromatin accessibility and spatially resolved transcriptomic data.

The combination of a variational autoencoder and optimal transport—two

frameworks that are becoming increasingly prominent in single-cell analysis—is interesting and the methods and results are well presented. Furthermore, the choice of datasets for comparison is entirely appropriate and varied. However, I think the reasoning for why combining the VAE and OT frameworks is useful need to be expanded upon significantly. In addition, the paper needs to consider and compare more recent multi-omic integration methods, as UOT has been used for integration of multi-omics data.

Response: We sincerely appreciate the highly insightful and constructive suggestions by Reviewer #2 so that we can improve the manuscript accordingly.

In the revised manuscript, we expanded the discussion of our proposed framework of combining VAE and OT, and compared it with other OT-based methods (see lines 315-336 on page 12).

We provided the detailed one-to-one responses below.

Major comments:

R2.1: Comparing uniPort to a wider range of multi-omics-based methods is critical. In particular, it would be worth considering: more recent methods in Seurat ([https://www.cell.com/cell/fulltext/S0092-8674\(21\)00583-3](https://www.cell.com/cell/fulltext/S0092-8674(21)00583-3)); scMC (<https://doi.org/10.1186/s13059-020-02238-2>); while the scVI-tools framework has a tool, totalVI, that is specifically for multi-omic integration (<https://doi.org/10.1038/s41592-020-01050-x>).

Response: We thank the reviewer for the valuable suggestion. We revised the manuscript by benchmarking more multi-omics methods (lines 114-165 on pages 5-6).

- We included scMC on two datasets of PBMC examples (Fig. 2; Supplementary Figs. 2, 3) and a dataset of the mouse spleen example (Figs. 3, 4), while we ignored scMC in MERFISH data as we encountered the problem of insufficient memory. scMC showed state-of-the-art performance on batch effect correction of one modality but has not been benchmarked on single-cell multi-omics data integration. Our result showed that scMC had similar performance with SCALEX in the PBMC example, which accurately preserved the data distribution. For the mouse spleen example, scMC achieved comparable results with uniPort.
- We also added GLUE and SCOT into comparison, which are the most advanced methods designed for single-cell multi-omics data integration, on the two PBMC (Fig. 2; Supplementary Figs. 2, 3) and the mouse spleen examples (Figs. 3, 4). We found that uniPort achieved comparable results with GLUE on the two PBMC examples and outperformed GLUE on the mouse spleen example. uniPort outperformed SCOT on all examples.
- In this study, we included the latest version of Seurat (v4.1.0) instead of using version 3. Fortunately, the updates from version 3 to version 4 are not relevant to our paper. The updates regard the processing of paired multi-omics datasets (different omics measured in the same cell), whereas our method and benchmarking and analysis are for the much more common situation of unpaired multi-omics datasets (different omics measured in different cells). Although one paired dataset was utilized, paired information was only used in the evaluation of performance, but not in training,

- Besides, to the best of our knowledge, totalVI in scvi-tools was specially designed

- Besides, to the best of our knowledge, totalVI in scvi-tools was specially designed for the integration of RNA and surface proteins with CITE-seq. Therefore, we also did not include totalVI in our comparison.

For details on scMC, SCOT, and GLUE implementations, please see lines 562-580 on pages 21-22, and for details on Seurat implementations, please see lines 517-532 on pages 19-20.

R2.2: It would be worthwhile to compare uniPort to other OT-based methods, such as SpaOTsc (<https://doi.org/10.1038/s41467-020-15968-5>) and SCOT (<http://doi.org/10.1089/cmb.2021.0446>), which are used for spatial deconvolution and multi-omics integration, respectively, as uniPort is the first method that I know of that uses mini-batch UOT for single-cell analysis.

Response: We thank the reviewer for the valuable comment and suggestions. In the revised manuscript, we made the following improvements:

- We benchmarked OT-based integration methods, SCOT, on the PBMC and mouse spleen examples (Figs. 2, 3, Supplementary Fig. 3). We found that the performance of SCOT is unstable as it integrated the paired PBMC dataset well with the third highest Silhouette coefficient and the second highest Batch Entropy score, but failed to align the mouse spleen dataset with low evaluated scores. For details on SCOT implementation, please check lines 574-580 on pages 21-22.
- We benchmarked uniPort against SpaOTsc and Tangram on a synthetic STARmap dataset from a recent benchmarking paper

(<https://doi.org/10.1038/s41592-022-01480-9>) (lines 213-233 on pages 8-9; Supplementary Fig. 7). Tangram is a global matrix optimization method, which aims to find a mapping matrix to project scRNA data to spot. SpaOTsc applies unbalanced and structured Gromov-Wasserstein optimal transport to find an optimal transport plan between scRNA-seq data and spot. We benchmarked the results of uniPort, Tangram, and SpaOTsc with four metrics: Pearson correlation coefficient (PCC), structural similarity index (SSIM), root mean square error (RMSE), and Jensen-Shannon divergence (JSD). Higher PCC and SSIM and lower RMSE and JSD, indicate better performance. We adopted the results of Tangram and SpaOTsc directly from the benchmarking paper. Overall, uniPort performed competitively with the two methods: uniPort performed favorably with PCC of 0.449, RMSE of 0.157, and JSD of 0.569, which is below Tangram (PCC of 0.619, RMSE of 0.147, and JSD of 0.524) while above SpaOTsc (PCC of 0.409, RMSE of 0.197, and JSD of 0.573) (Supplementary Fig. 7).

Supplementary Figure 7. Spatial deconvolution benchmark. Pearson correlation coefficient (PCC), structural similarity index (SSIM), root mean square error (RMSE) and Jensen-Shannon divergence (JSD) of uniPort, Tangram and SpaOTsc over simulated STARmap example. Higher PCC and SSIM, and lower RMSE and JSD, indicate better performance.

R2.3: The authors state that minibatch OT is used to allow uniPort to be applied to larger datasets, which are common in single-cell analysis. However, I do not see any comparison of runtime between the different considered integration frameworks. I think that would help strengthen the methods benchmarking.

Response: We thank the reviewer for the valuable suggestion. We added a runtime benchmark on the paired PBMC example in Supplementary Fig. 11. We found that uniPort has comparable runtime with VAE-based models, e.g., scVI, SCALEX, and GLUE, and consumes much less computational time than those by global optimal transport-based methods, e.g., SCOT and Pamona (10+ minutes vs. 100+ minutes for 10K cells).

Supplementary Figure 11. Running time of part of compared methods on PBMC example.

R2.4: One of the proposed features of uniPort is online prediction after constructing a reference atlas, which is demonstrated by predicting scRNA-seq gene expression onto spatial MERFISH data, which has significantly lower gene coverage. I think comparing uniPort to other methods specialized for mapping scRNA-seq gene expression onto spatial data, such as Tangram or gimVI, which were considered in a recent benchmarking study (<https://doi.org/10.1038/s41592-022-01480-9>).

Response: We thank the reviewer for the valuable suggestions. Our online gene prediction for mouse #2 is derived from MERFISH data of mouse #2 and a pre-trained network by data of mouse #1. This is based on the feature of the generative model. However, Tangram used a cell-to-cell probabilistic matching matrix between scRNA and MERFISH, which can only be used for imputing genes for training MERFISH data of mouse #1, rather than MERFISH data from another mouse #2.

Therefore, to better benchmark uniPort against Tangram and gimVI, we further added a new task (lines 184-207 on pages 7-8) which included both Tangram and gimVI for performance comparisons. To be specific, we have added the following comparison.

- We followed the scheme of gimVI to impute missing genes for MERFISH. we first randomly selected 80% (i.e., 122/153) genes in MERFISH as training genes and reserved the remaining 20% (i.e., 31/153) genes as testing genes. We repeated the above steps twelve times and obtained 12 training and testing gene sets. Afterwards, we trained the uniPort network with each training gene set and then imputed the corresponding testing gene set.

- We compared our results with gimVI and Tangram

- We compared our results with gimVI and Tangram.
We applied uniPort, gimVI, and Tangram to impute testing genes, and used UMAP to visualize both training and testing genes (Fig. 5). With an imputation framework like that of gimVI, we also expected imputed values of uniPort to carry gene-specific biases from scRNA genes. Therefore, for performance evaluation, we followed gimVI and reported the median and average Spearman correlation coefficients (mSCC and aSCC), as well as the median and average Pearson correlation coefficients (mPCC and aPCC) over imputed and ground truth testing genes. uniPort provided a significant improvement over the two compared methods on the MERFISH dataset. For example, uniPort separated different cell types in the UMAP visualization of imputed genes with a better result (Fig. 5e), and demonstrated the highest mSCC (0.259), aSCC (0.26), mPCC (0.249), and aPCC (0.294), significantly above those of gimVI (mSCC of 0.221, aSCC of 0.24, mPCC of 0.201, and aPCC of 0.242) and Tangram (mSCC of 0.188, aSCC of 0.206, mPCC of 0.202, and aPCC of 0.231) (Fig. 5f).

Fig. 5: **e**, UMAP visualization of imputed MERFISH genes of Tangram, gimVI and uniPort.; **f**, Median and average Spearman correlation coefficient (SCC) and Pearson correlation coefficient (PCC) between real and imputed MERFISH genes.

R2.5: For Minibatch-UOT, the parameter ρ related to marginal penalization in equation (4) plays an important role on the integration, especially when there are rare cell types that are present in only one dataset. What is the reason to set $\rho=1$? When the value of ρ is changed, can uniPort still separate Ependymal cells, which

are only observed in MERFISH data? Do the results of Figure 3a change significantly?

Response: We thank the reviewer for the robustness of uniPort on varying ρ . In the revised manuscript, we calculated the values of F1, NMI, ARI, Batch Entropy score, and Silhouette coefficients for uniPort with different choices of ρ values ($\rho=0.0625, 0.125, 0.25, 0.5, 1, 2, 4, 8, 16$) for both cell type balanced paired PBMC and cell type unbalanced MERFISH data (Supplementary Fig. 17c). We found that uniPort achieved robust performance under different choices of ρ .

Supplementary Fig. 17: **c**, evaluated scores of uniPort for different choices of ρ over the paired PBMC and MERFISH example.

We also visualized the results of MERFISH integration in the cases of $\rho=0.125$ and $\rho=8$, respectively. uniPort still accurately separated ependymal cells of MERFISH data in the two cases, with negligible alterations on the data visualization structures shown below (figures not shown in the manuscript).

R2.6: The input and output of uniport is unclear, e.g., XC, XS... in Figure 1, as the notation for input and output is the same. Similarly, in the Methods section, uniPort also requires XS, YS as input, suggesting that uniPort does not generate data-specific genes but rather requires them as input.

Response: We thank the reviewer for pointing out this issue.

In the revised version, we clarified our representation by modifying the output X_C , X_S

in the revised version, we clarified our representation by modifying the output X^S , X^C , Y^C , Y^S as \hat{X}^C , \hat{X}^S , \hat{Y}^C , \hat{Y}^S to distinguish the input and reconstruction data in Fig. 1b. We also clarified this in the corresponding main text (lines 411-414, on page 16).

When training the model, uniPort does need X^S and Y^S as training data. However, after model training, we can use the trained network to impute data-specific genes by common genes.

Fig. 1: **b**, uniPort projects input datasets into a cell-embedding latent space through a shared probabilistic encoder. Then uniPort minimizes a Minibatch-UOT loss between cell embeddings across different datasets. Finally, uniPort reconstructs two terms. The first consists of input datasets by a decoder with different DSBN layers. The second consists of highly variable gene sets corresponding to each dataset by dataset-specific decoders.

R2.7: To predict genes in scRNA-seq data by the trained reference atlas (Line 197), only 153 common genes are used as input. How do the authors consider and deal with possible batch effects between samples?

Response: We thank the reviewer for pointing out the existence of possible batch effects. We agree with the reviewer that imputed values by uniPort or other methods (e.g., gimVI) may carry batch effects between samples, which was also pointed out by gimVI [see page 4 of gimVI paper (<https://romain-lopez.github.io/publication/gim-vi>)]. To deal with possible batch effects during performance evaluation, we, therefore, followed gimVI to consider using the Pearson/Spearman correlation coefficients between imputed and ground truth gene values, instead of using mean square errors of the imputed absolute values (see lines 198-207 on page 8).

R2.8: When using evaluation metrics for benchmarking, I think it would help that the authors were clearer about the comparison. For example, are the authors evaluating known clustering vs predicted clustering? For the single-cell RNA-sequencing and single-cell ATAC-sequencing data of peripheral mononuclear blood cells, do the authors consider the consistency of paired single cell labels?

Response: We thank the reviewer for pointing out this issue. In the revised version, we clarified the Evaluation metrics in the main text with more details added as follows:

- Add details about calculating evaluation metrics

We added details about how to calculate the evaluated F1, ARI, and NMI scores through predicted and real cell type annotations (lines 593-599 on page 22):

“To be specific, we trained a k-Nearest-Neighbor (kNN) classifier by the sklearn.neighbors.KNeighborsClassifier function based on cell type annotations

sklearn.neighbors.KNeighborsClassifier function based on cell-type annotations and UMAP coordinates of reference data (e.g., scRNA) embeddings in the common latent space. Then, we applied the well-trained kNN classifier to predict cell-type annotations of query dataset (e.g., scATAC/MERFISH) embeddings, and calculated the ARI, NMI and F1 scores by real and predicted query cell type annotations.”

- Add the average FOSCTTM to consider the consistency of the paired cells.
For the paired PBMC example, we added the average FOSCTTM score as another reviewer suggested, to test the consistency of paired cells. FOSCTTM refers to the “fraction of samples closer than the true match”. It is used to measure the preservation of cell-cell correspondence across datasets. We evaluated the average FOSCTTM on the paired PBMC example in Fig. 2e. We found that uniPort achieved an average FOSCTTM of 0.0694, ranked second among 11 compared methods, slightly below GLUE. For more details about the average FOSCTTM (see lines 624-630 on page 24).

Minor comments:

R2.9: Please double check the figure captions. For example, in Figure 3a, only the integration results from uniPort are shown, rather than the results from Harmony, Seurat, SCALEX or scVI.

Response: We thank the reviewer for pointing out these typos. In the revised manuscript, we checked the figure captions. We corrected the caption of the original Figure 3a by removing ‘Harmony, Seurat, SCALEX and scVI’ (now Fig. 5).

R2.10: It is good to see the comparison between gene expressions in MERFISH and predicted RNA as in Supplementary Fig 7. What about the real single cell RNA expression? It seems like the authors only show the comparison the mean expressions between real and predicted RNA.

Response: We thank the reviewer for the valuable comment.

We did not have paired cell-to-cell scRNA data for MERFISH profiled from mouse #2. Therefore, we can only calculate the correlation between mean expressions instead of the same samples.

We trained the MERFISH data from mouse #1 with scRNA data from other mice and predicted scRNA genes through MERFISH data from mouse #2. These predicted scRNA genes are used to enhance the resolution of MERFISH genes of mouse #2, as MERFISH can only measure a small number of genes.

Besides, we also performed another imputation task as stated in **R2.4**. The result also demonstrated the uniPort's ability for gene imputation.

R2.11: In the Supplementary information, line 55: 'bisaed', which I think should be 'biased'. I think this is the only spelling error, but please do check over the rest of the manuscript to make sure.

Response: Thank you for pointing out this typo. In the revised supplementary, we corrected this typo 'biased'.

corrected this typo. b1saed.

REVIEWERS' COMMENTS

Reviewer #1 (Remarks to the Author):

The authors have thoroughly addressed my questions.

Reviewer #2 (Remarks to the Author):

The authors have addressed my comments.

Responses to Reviewers:

We would like to thank the reviewers for their valuable time and efforts on our manuscript. The point-by-point responses to the reviewers' comments and suggestions are as follows and our responses are in red color.

Reviewer #1:

The authors have thoroughly addressed my questions.

Response: We sincerely thank the reviewer for the comment.

Reviewer #2:

The authors have addressed my comments.

Response: We sincerely thank the reviewer for the comment.